# Comparison of tropospheric $NO_2$ columns from MAX-DOAS retrievals and regional air quality model simulations

Anne-Marlene Blechschmidt[1], Joaquim Arteta[2], Adriana Coman[3], Lyana Curier[4,*], Henk Eskes[5], Gilles Foret[3], Clio Gielen[6], Francois Hendrick[6], Virginie Marécal[2], Frédérik Meleux[7], Jonathan Parmentier[2], Enno Peters[1], Gaia Pinardi[6], Ankie J. M. Piters[5], Matthieu Plu[2], Andreas Richter[1], Arjo Segers[4], Mikhail Sofiev[8], Álvaro M. Valdebenito[9], Michel Van Roozendael[6], Julius Vira[8], Tim Vlemmix[10], and John P. Burrows[1]

[1]Institute of Environmental Physics, University of Bremen, IUP-UB, Bremen, Germany
[2]Centre National de Recherches Météorologiques, Météo-France-CNRS, UMR 3589, Toulouse, France
[3]Laboratoire Interuniversitaire des Systèmes Atmosphériques, CNRS/INSU UMR7583, Université Paris-Est Créteil et Université Paris Diderot, Institut Pierre Simon Laplace, Créteil, France
[4]TNO, Climate Air and Sustainability Unit, Utrecht, the Netherlands
[5]Royal Netherlands Meteorological Institute, KNMI, De Bilt, the Netherlands
[6]Royal Belgian Institute for Space Aeronomy, BIRA-IASB, Brussels, Belgium
[7]Institut National de l'Environnement et des RISques industriels, INERIS, Verneuil en Halatte, France
[8]Finnish Meteorological Institute, FMI, Helsinki, Finland
[9]Norwegian Meteorological Institute, MetNo, Oslo, Norway
[10]TU-Delft, Delft, the Netherlands
[*]now at: Faculty of Humanities and Sciences, Department of Biobased Materials, Maastricht University, Geleen, the Netherlands

*Correspondence to:* A.-M. Blechschmidt (anne.blechschmidt@iup.physik.uni-bremen.de)

**Abstract.** Tropospheric NOx (NO+$NO_2$) is hazardous to human health and can lead to tropospheric ozone formation, eutrophication of ecosystems and acid rain production. It is therefore important to establish accurate data based on models and observations to understand and monitor tropospheric $NO_2$ concentrations on a regional and global scale.

In the present study, MAX-DOAS tropospheric $NO_2$ column retrievals from four European measurement stations are compared to simulations from five regional air quality models which contribute to the European regional ensemble forecasts and reanalyses of the operational Copernicus Atmosphere Monitoring Service (CAMS). Compared to other observational data usually applied for regional model evaluation, MAX-DOAS data is closer to the regional model data in terms of horizontal and vertical resolution and multiple measurements are available during daylight, so that for example diurnal cycles of trace gases can be investigated.

In general, there is good agreement between simulated and retrieved $NO_2$ column values for individual MAX-DOAS measurements with correlations roughly between 35 and 70 % for individual models and 45 to 75 % for the ensemble median for tropospheric $NO_2$ VCDs, indicating that emissions, transport and tropospheric chemistry of NOx are on average well simulated. However, large differences are found for individual pollution plumes observed by MAX-DOAS. Most of the models overestimate seasonal cycles for the majority of MAX-DOAS sites investigated. At the urban stations, weekly cycles are reproduced well but the decrease towards the weekend is underestimated and diurnal cycles are overall not well represented. In

particular, simulated morning rush hour peaks are not confirmed by MAX-DOAS retrievals and models fail to reproduce observed changes in diurnal cycles for weekdays versus weekend. A large number of evaluation points arise from the comparison to MAX-DOAS measurements which should be used in future regional air quality modelling studies to track down reasons of disagreement.

## 1 Introduction

Nitrogen dioxide ($NO_2$) is a key species for atmospheric chemistry. Photolysis of $NO_2$ leads to formation of tropospheric ozone. The latter is a major greenhouse gas and the main precursor of OH, which itself determines the oxidising capacity of the atmosphere. Oxidation to $HNO_3$ via reaction with OH (daytime) or ozone (nighttime) is the major sink of $NO_2$ in the troposphere (Jacob, 1999) and results in acid rain and eutrophication of ecosystems, which are both harmful for the environment. Moreover, $NO_2$ can cause irritation of respiratory organs (http://www3.epa.gov/).

Within the troposphere, conversion of NO to $NO_2$ only takes about a minute during daytime. The sum of NO and $NO_2$ is called NOx, which is mainly emitted in the form of NO to the atmosphere. Main sources of NOx are fossil fuel combustion and biomass burning. Some NOx is also produced from lightning and microbial activity in soils.

The lifetime of NOx is only a few hours in the boundary layer but a few days in the upper troposphere, where less OH radicals are present (Ehhalt et al., 1992) to react with $NO_2$ and more NOx is present as NO which has fewer permanent sinks than $NO_2$. Several studies (e.g. Stohl et al., 2003; Zien et al., 2014) have shown that in the free troposphere, $NO_2$ can be transported over larger distances and is hence not only important for regional but also for global air quality. Peroxyacyl nitrate (PAN) produced by photochemical oxidation of carbonyl compounds is not much affected by wet scavenging and can act as a reservoir of $NO_2$, especially during long-range transport. If the air masses descend away from their source regions, PAN will decompose to NOx under the influence of, on average, higher temperatures at lower altitudes (Jacob, 1999).

Given the influence of NOx on air quality and climate through effects on radiation, it is of high environmental and scientific interest to accurately observe and simulate spatial distribution and time evolution of $NO_2$ concentrations in the troposphere. Simulating $NO_2$ is a challenge for numerical models as it is chemically very active and depends on many factors including for example cloud cover which affects photolysis of this trace gas. Moreover, representation of NOx emissions adds a large uncertainty to the model output.

MAX-DOAS (Multi Axis Differential Optical Absorption Spectroscopy; e.g. Hönninger et al., 2004; Wittrock et al., 2004) measurements have been used to investigate air pollution in many studies, including the FORMAT campaign in Northern Italy (Heckel et al., 2005; Wagner et al., 2011), the CINDI campaign in the Netherlands (Piters et al., 2012), campaigns in Canada (Halla et al., 2011; Mendolia et al., 2013), China (e.g. Irie et al., 2011; Hendrick et al., 2014; Ma et al., 2013; Wang et al., 2014), during ship-borne measurements (Leser et al., 2003; Takashima et al., 2012; Peters et al., 2012).

MAX-DOAS observations of atmospheric composition are performed by taking measurements of the scattered sunlight at different elevation and sometimes also azimuthal angles. Depending on the viewing angle and solar position, the light path through the atmosphere is different, with the observation in the zenith direction usually providing the shortest light path through

the lower troposphere. Therefore, using observations in low elevation angles as measurement intensity and zenith measurements as reference intensity, the total amount of molecules of a certain species along the light path difference (zenith subtracted from non-zenith measurement), so called differential slant column densities, can be determined using Lambert Beer's law. These can be inverted to tropospheric columns and lower altitude tropospheric profiles by radiative transfer modelling and optimal

estimation techniques.

A large number of studies applied MAX-DOAS data for satellite validation (e.g. Celarier et al., 2008; Valks et al., 2011; Irie et al., 2008; Irie et al., 2012; Ma et al., 2013; Lin et al., 2014; Kanaya et al., 2014; Pinardi et al., 2014) but up to now, comparisons to regional air quality model simulations of tropospheric $NO_2$ have, to our knowledge, only been carried out by Vlemmix et al. (2015) and Shaiganfar et al. (2015). Several studies compared regional air quality model simulations to

satellite data (e.g. Huijnen et al., 2010), although satellite data are usually only available at much coarser time steps compared to regional model data. In this respect, the advantage of MAX-DOAS retrievals compared to satellite retrievals is the high resolution in time. Moreover, several studies compared in-situ $NO_2$ data to regional model results (e.g. Vautard et al., 2009; Colette et al., 2011; Mues et al., 2014), although in-situ data usually refer to a specific location (point measurements), whereas regional model results are available for a specific horizontal grid resolution and area depending on the model set up. As MAX-

DOAS data represents a larger volume of air, it is much better suited for investigating performance of regional models than in-situ data. According to Richter et al. (2013) the horizontal averaging volume of MAX-DOAS data depends on aerosol loading, wavelength and viewing direction and ranges from a few kilometres in the polluted boundary layer up to 80 km from the top of a mountain under clean air conditions. Another advantage of MAX-DOAS measurements is their ability to observe several pollution related species at the same time (e.g. $NO_2$, HCHO, CHOCHO, $SO_2$, aerosols, potentially also $O_3$) and to

provide $NO_2$ data which is virtually free of interferences from other species or nitrogen compounds such as NOy (NOx and other oxidised nitrogen species). In contrast to $NO_2$, NOx cannot be retrieved from MAX-DOAS measurements directly, so that these measurements are of more interest for air quality than for atmospheric chemistry studies. Vertical profiles of trace gases can be retrieved from MAX-DOAS measurements, which is another advantage for model comparison studies.

In the present study, MAX-DOAS measurements are compared to regional air quality model simulations to investigate

model performance. Parts of this approach are already applied within scientific reports of the operational Copernicus Atmosphere Monitoring Service (CAMS, http://atmosphere.copernicus.eu/), see e.g. Blechschmidt et al. (2015) and Eskes et al. (2018), but mainly to model results provided on 8 output levels only, which introduces uncertainty to comparison results. CAMS is the operational follow-up of the former GEMS (Global and regional Earth-system Monitoring using Satellite and in-situ data) (Hollingsworth et al., 2008) and three succeeding MACC (Monitoring Atmospheric Composition and Climate,

http://www.gmes-atmosphere.eu/) projects. The global component of CAMS extends weather services of the ECMWF (European Centre for Medium-Range Weather Forecasts) with simulations of atmospheric trace gases and aerosols, while operational air quality forecasts and analyses for Europe are provided at much higher resolution through the regional component. Hourly $NO_2$ vertical column densities (VCDs) from 6 different regional model runs based on 5 models which are used within CAMS will be compared to MAX-DOAS measurements from three urban and one rural European station: Bremen (operated

by IUP-Bremen), De Bilt (operated by KNMI), Uccle and OHP (Observatoire de Haute-Provence) (the latter two operated by

BIRA-IASB). Location of the stations are plotted on top of mean $NO_2$ tropospheric columns from OMI (Levelt et al., 2006) satellite observations for February 2011 as well as on a map of anthropogenic NOx emissions used by the models in Figure 1 as an indicator of pollution levels in these and surrounding regions. The spatial distribution of NOx emissions agrees well with pollution hotpots and cleaner areas identified by OMI.

Due to the large number of model evaluation points arising from the MAX-DOAS based comparisons, the reasons for differences between model results and observations found by the comparisons are discussed here in a general sense and need to be further investigated e.g. by carrying out additional dedicated model runs in future modelling studies.

The manuscript starts with an overview of regional model and MAX-DOAS data (Section 2) followed by a description of the comparison method (Section 3). Results are described and discussed in Section 4. Finally, a summary and conclusions are
given in Section 5.

## 2   Data basis

### 2.1   Regional air quality model simulations

CHIMERE (Menut et al., 2013), LOTOS-EUROS (LOng Term Ozone Simulation - EURopean Operational Smog) (Schaap et al., 2008), EMEP MSC-W (European Monitoring and Evaluation Programme Meteorological Synthesizing Centre - West)
(Simpson et al., 2012), MOCAGE (Model Of atmospheric Chemistry At larGE scale) (Josse et al., 2004; Guth et al., 2016) and SILAM (System for Integrated modeLling of Atmospheric coMposition) (Sofiev et al., 2006; Sofiev et al., 2015) contributed to the European regional ensemble forecasts (Marécal et al., 2015) and reanalyses of the former MACC projects and are currently used within CAMS. These models have been used in many studies for investigating atmospheric composition on a regional scale (e.g. Drobinski et al., 2007; Huijnen et al., 2010; Lacressonnière et al., 2014; Petetin et al., 2015; Solazzo et al., 2012;
Watson et al., 2016; Zyryanov et al., 2012).

All of these models use ECMWF-IFS and MACC reanalysis (Innes et al., 2013) data as meteorological and chemical input data and boundary conditions, respectively. Anthropogenic emissions are taken from the MACC emissions database (Kuenen et al., 2011), GFAS (Kaiser et al., 2012) is used to account for fire emissions. The input to these models is thus consistent and hence, differences in model results are due to differences in the modelling code, model set up or due to different scalings
of emissions e.g. to account for seasonal, diurnal and weekly cycles as well as emission heights. The model runs investigated in the present study were performed by different European institutions and are based on different horizontal and vertical grid spacings and chemistry schemes (see Table 1 for further details). Apart from SILAM, the models were run without chemical data assimilation. The SILAM simulations included assimilation of surface observations of $NO_2$ as described in Vira and Sofiev (2015).
Two different sets of EMEP model runs are investigated in this study. The first one uses the same setup as the other regional models described above and is termed EMEP-MACCEVA in the following. EVA (validated assessments for air quality in Europe) was a subproject of MACC dedicated to the development and implementation of operational yearly production of European air quality assessment reports (https://www.gmes-atmosphere.eu). The second set of simulations (called EMEP in

the following) uses the same set-up as in the EMEP status reports (see http://www.emep.int) for each year based on the EMEP subdomain, ECMWF-IFS as meteorological driver, EMEP emissions, Fire INventory from NCAR version 1.0 (FINNv1; Wiedinmyer et al., 2011), initial conditions described by Schulz et al. (2013) for the years 2010-2011 and Fagerli et al. (2014) for 2012 and climatological boundary conditions described by Simpson et al. (2012).

According to Mues et al. (2014), chemistry transport models in general account for seasonal, daily and diurnal emission changes by applying average time profiles given for different energy sectors and regions to totals of annual emissions across the model domain. Temporal emission patterns used by the regional air quality models listed above are country and SNAP (Selected Nomenclature for Sources of Air Pollution) sector dependent and are based on Denier van der Gon et al. (2011). A list of the SNAP sectors is given by Bieser et al. (2011). Moreover, different vertical emission profiles are applied for each

regional model. These are described in more detail by Bieser et al. (2011) for EMEP and CHIMERE, Simpson et al. (2003) for SILAM and Thunis et al. (2010) for LOTOS-EUROS. For MOCAGE, emissions are injected into the five lowest model levels using a hyperbolic decay.

More details on specific model setups and scores with respect to surface observations, can be found in Marécal et al. (2015) and in the model specification/validation dossiers which are available online at:

http://www.gmes-atmosphere.eu/about/documentation/regional/.

## 2.2   MAX-DOAS retrievals

This study makes use of MAX-DOAS measurements from four European stations: Bremen (Germany), De Bilt (the Netherlands), Uccle (Belgium), and OHP (France). Characteristics of the data available from the stations, such as exact location and time period of retrievals investigated here, are briefly summarized in Table 2 and will be described below.

For Bremen, Uccle and OHP, $NO_2$ slant column densities (SCDs) are obtained by a DOAS analysis for a specific wavelength window using a series of low elevation angles as measurement intensity and zenith measurements as reference intensity. Cross sections of different trace gases are accounted for in the retrieval. Resulting SCDs of $NO_2$ and $O_4$ are then used as input for a radiative transfer model which is a two-step approach. First, an aerosol extinction profile is retrieved by comparing the measured $O_4$ SCDs to $O_4$ SCDs simulated by the radiative transfer models SCIATRAN (Rozanov et al., 2005) for Bremen and

bePRO (Clémer et al., 2010) for Uccle and OHP. In the second step, the derived aerosol extinction, measured $NO_2$ SCDs and an a-priori $NO_2$ profile are used to retrieve the $NO_2$ profile of interest. This is an inverse problem solved by means of the optimal estimation method (Rodgers , 2000). The Maxdoas Retrieval algorithm of KNMI (MARK) uses a least squares minimization of the differences between measured and modeled differential slant column densities, by interpolation of look-up tables. The look-up tables are calculated with the radiative transfer model DAK (Doubling Adding KNMI; De Haan et al., 1987; Stammes ,

2001). With this method, a maximum of four parameters are retrieved, which together determine the profile shape: tropospheric vertical column, boundary layer height, gradient in the boundary layer, fraction of $NO_2$ in the free troposphere.

De Bilt (52.10° N, 5.18° E; see Figure 1) is the home town of KNMI, and located just outside the city of Utrecht. The De Bilt experimental research site is surrounded by local and regional roads, with a lot of traffic which can affect regional air quality significantly. According to Vlemmix et al. (2015), it can also be affected by pollution sources which are located more

far away in the Rotterdam region to the south-west, Amsterdam to the north-west and the German Ruhr region to the south-east of De Bilt. The MAX-DOAS instrument operated at De Bilt is a commercial system obtained from Hoffmann Messtechnik. It has an Ocean Optics spectrograph, diffraction grating and a CCD detector. It operates at a wavelength range of 400-600 nm. The pointing direction of the instrument is 80° (east to north-east), the wavelength window of the DOAS fit for $NO_2$ is 425-490nm. Wavelength calibration and slit-function width are determined using a high-resolution solar spectrum. Cross sections of $O_3$, $NO_2$, $O_4$, $H_2O$ and a pseudo cross-section accounting for the Ring effect are applied. The choice of fitting parameters complies with the standards agreed by the MAX-DOAS community, following from homogenization efforts within e.g. CINDI, GEOMON, NORS and QA4ECV as much as possible. Air mass factor (AMF) calculations are performed by the DAK model. A-Priori profiles of $NO_2$ are based on a block-profile with $NO_2$ present in the boundary layer, boundary layer heights were taken from a climatology based on ECMWF data. For De Bilt, averaging kernels refer to the altitude-dependent (or box-)differential AMFs divided by the total differential AMF. The differential AMF is derived at a specific altitude by simulating the radiance with and without an added partial column of $NO_2$ at this altitude with the DAK radiative transfer model. $NO_2$ columns are retrieved from the measurements at De Bilt, $NO_2$ profiles are not available.

The IUP-Bremen MAX-DOAS instrument consists of an outdoor telescope unit collecting light in different directions, and an indoor grating spectrometer (Shamrock 163 equipped with an Andor LOT257U CCD with 2048x512 pixels) covering a wavelength interval from 430–516 nm at a resolution of approximately 0.7 nm. Both components are connected via an optical fiber bundle which simplifies handling and overcomes polarization effects. The telescope unit is installed at an altitude of approximately 20 m above ground level at the roof of the Institute of Environmental Physics building (53.11° N, 8.86° E) at the University of Bremen which is located to the north-east of the city centre. The azimuthal pointing direction is north-west, which means that some of the measured pollution peaks are due to the exhaust of an industrial area, predominantly a steel plant, as well as a near-by highway. However, over longer time periods the retrievals should be dominated by pollution from the city centre. $NO_2$ SCDs are derived by DOAS analysis using a fitting window of 450-497 nm. Cross sections of $O_3$, $NO_2$, $O_4$, $H_2O$ and a pseudo cross-section accounting for the Ring effect are accounted for in the fit. Profiles of $NO_2$ are derived from SCDs applying the BRemian Advanced MAX-DOAS retrieval algorithm (BREAM) which incorporates SCIATRAN radiative transfer simulations. An $NO_2$ a-priori which is constant with altitude is assumed and iterated in the retrieval. Detailed information about the profile retrieval is given by Wittrock et al. (2006) and Peters et al. (2012).

BIRA-IASB operates a MAX-DOAS instrument at OHP (Observatoire de Haute-Provence; 43.92° N, 5.7° E) since 2005. OHP is a background remote site in the south of France, temporarily affected by transport of pollution from regional sources (e.g. from the petrochemical plants of Etang de Berre close to Marseille in the south-west) and the Po valley (Italy) to the north-east of the station. The MAX-DOAS instrument, which points towards the SSW direction, consists of a grating spectrometer Jobin-Yvon Triax 180 (1800 grooves/mm) covering the 330-390nm wavelength range coupled to a thermo-electrically cooled (-40°C) Hamamatsu CCD detector (1024 pixels). $NO_2$ SCDs are obtained by applying the DOAS technique to a 364-384 nm wavelength interval, taking into account spectral signatures of $O_3$, $O_4$, the Ring effect and $NO_2$. At Uccle (50.8° N, 4.32° E), which is located south-west of the Brussels city centre, a mini-MAX-DOAS from Hoffmann Messtechnik GmbH covering the 290-435 nm wavelength range is operated by BIRA-IASB since 2011. The instrument is pointing north to north-east towards

the city centre. NO$_2$ SCDs are retrieved in a 407-432 wavelength interval including the same spectral signatures as for OHP. It should be noted that a sequential zenith reference spectrum has been implemented in order to minimise the impact of changes in shift and resolution due to temperature instabilities. The DOAS fit for NO$_2$ has also been improved by introducing pseudo-absorber cross-sections derived from principal component analysis of residuals on days affected by large thermal instabilities.

This approach allows for a better correction of fast-changing slit-function variations, resulting in more stable residuals and therefore more realistic random uncertainty estimates. For NO$_2$ vertical profile retrievals at both stations, the bePRO radiative transfer code (Clémer et al., 2010) is used. NO$_2$ profiles are retrieved at 420 nm for Uccle and 372 nm for OHP. For NO$_2$ vertical profile retrievals, exponentially decreasing a-priori profiles have been constructed, based on an estimation of NO$_2$ vertical column densities derived from the so-called geometrical approximation (Hönninger et al., 2004; Brinksma et al., 2008)

and using scaling heights of 0.5 and 1 km for OHP and Uccle, respectively. A-priori and measurement-uncertainty covariance matrices are constructed as by Clémer et al. (2010) with adopted correlation lengths of 0.05, and covariance scaling values of 0.5 and 0.35 for Uccle and OHP, respectively. For this study, only retrievals with a residual of the optimal estimation method retrieval fit to the DSCDs smaller than 50 % and degrees of freedom for signal larger than 1 are used. A more detailed description of the model and trace gas profile retrievals can be found in Hendrick et al. (2014). Although there has not been

formal side-by-side operation of both instruments for verification purpose, a good overall agreement has been obtained between the mini-DOAS and other BIRA research-grade spectrometers similar to the one operated at OHP, e.g. like during the CINDI campaign (Roscoe et al., 2010).

Previous studies (e.g. Hendrick et al., 2014; Wang et al., 2014; Franco et al., 2015) have shown that the typical error on MAX-DOAS retrieved VCDs is around 20 %, including uncertainties related to the optimal estimation method, trace gas cross

sections and aerosol retrievals, and can be higher for sites with low trace gas concentrations like OHP or due to instrumental conditions. Moreover, the uncertainty of the retrieval is increased in cloudy conditions.

For Uccle, information on cloud conditions was retrieved according to the method by Gielen et al. (2014) which is based on analysis of the MAX-DOAS retrievals, but not applied for results shown in the present study. No cloud flags are available for Bremen, De Bilt and OHP. Larger uncertainties are associated with retrievals under cloudy conditions in particular as clouds are

not included in the MAX-DOAS forward calculations. However, MAX-DOAS retrievals are usually filtered for patchy cloud situations by comparing radiative forward calculations of O$_4$ to retrieved O$_4$ columns and removing cases from the data with larger than expected differences. The presence of clouds may alter MAX-DOAS retrievals in several ways: (1) If clouds are present at both zenith and horizon viewing directions, NO$_2$ within and above the clouds is shielded from the MAX-DOAS view whereas the sensitivity is slightly increased below the cloud, (2) if a cloud is present at the zenith/non-zenith viewing direction

only, the sensitivity is reduced/enhanced at the height of the cloud and slightly enhanced/reduced below the cloud compared to the cloud free case. The impact of clouds on MAX-DOAS retrievals is described in detail by Vlemmix et al. (2015). In addition to the direct effect of clouds on the measurements, clouds also affect photolysis rates and hence NOx chemistry and NO to NO$_2$ partitioning, which may have an impact on tropospheric NO$_2$ columns and profiles retrieved under cloudy weather conditions. The influence of clouds on comparison results is hence complex and regarded as a topic for future studies.

## 2.3 Wind measurements

In order to investigate the ability of the models to reproduce transport of $NO_2$ towards the stations, the MAX-DOAS data described above is complemented by meteorological in-situ station data of wind speed and wind direction. Wind data for Bremen was provided by the German Weather Service/ Deutscher Wetterdienst through their website at http://www.dwd.de.
The weather station in Bremen is located at the main airport, approximately 9 km southwards of the MAX-DOAS station. This may result in some differences to the actual wind direction and wind speed at time and location of the MAX-DOAS retrievals. Wind data for OHP was taken from the weather station at the observatory and downloaded from the corresponding website at http://pc-meteo.obs-hp.fr/intervalle.php. Wind speed and direction measurements at Uccle are performed using a commercial rugged wind sensor from Young (model 05103) and were provided by BIRA-IASB through their webpage at
http://uvindex.aeronomie.be. For De Bilt, wind measurements (within 300 m from the MAX-DOAS instrument) carried out by KNMI were downloaded from https://www.knmi.nl/nederland-nu/klimatologie/uurgegevens.

## 3   Methodology for regional model evaluation

The sensitivity of MAX-DOAS retrievals is largest in the boundary layer, which needs to be taken into account when comparing MAX-DOAS retrievals to model simulated values. This is achieved here, by applying column averaging kernels (AVKs) to the
model data prior to comparison. The AVKs are part of the MAX-DOAS profiling output and represent the sensitivity of the retrieved column to the amount of $NO_2$ at different altitudes. Note that no profile data is available for De Bilt and AVKs were derived based on (box-)differential AMFs at that station (see Section 2.2).

In this study, model VCDs are derived by two different methods in order to test the influence of AVKs on the data analysis. Non AVK-weighted model VCDs are calculated by simply summing up $NO_2$ partial columns ($VCD_i$) over all N model levels
in the vertical:

$$VCD^{model}_{nonAVK-weighted} = \sum_{i=1}^{N model} VCD^{model}_i \qquad (1)$$

In addition, model VCDs are calculated by applying column AVKs of the retrievals to model $NO_2$ partial columns before summing up $NO_2$ partial columns in the vertical. The following data processing steps were carried out prior to the application of column AVKs:

(1) Conversion of provided model $NO_2$ partial columns [molec cm$^{-2}$] to concentrations [molec cm$^{-3}$] using model layer thicknesses.

(2) Deriving model concentrations on measurement altitudes assuming that model concentrations are constant within a specific model layer. If a measurement layer overlaps with more than one model layer, the result is a weighted mean over the model layer concentrations. If the highest measurement altitude is above the model top, the concentration at the model top level
is used. It is assumed here that the latter has no significant impact on the data analysis, as $NO_2$ concentrations are in general small towards higher elevation levels compared to lower levels.

(3) Conversion of derived $NO_2$ concentrations on measurement altitudes to partial columns [molec cm$^{-2}$] using observation layer thicknesses.

AVK-weighted model VCDs were then calculated using the following equation:

$$VCD_{AVK-weighted}^{model} = \sum_{i=1}^{Nobs} AVK_i * VCD_i^{model} \tag{2}$$

5     where Nobs is the number of measurement altitudes.

Note that non AVK-weighted and AVK-weighted model VCDs are based on the model output at original vertical resolution. VCDs are calculated separately for each model and constitute the basis for calculating ensemble mean values which are described at the end of this Section.

Only those model values closest to the measurement time are used below. As the model output is given in hourly time steps, 10  the maximum possible time difference between measurements and simulations is 30 minutes.

Following studies by e.g. Marécal et al. (2015), Langner et al. (2012), Solazzo et al. (2012), Vautard et al. (2009), the present manuscript focuses on results of the model ensemble, i.e. the median of individual model results of a given quantity. As an even number of 6 different model runs (based on 5 different models) constitute the model ensemble in the present study, the median is calculated by ordering the 6 different model values (e.g. for seasonal cycles, these values refer to the average of individual 15  model runs for each month) in terms of magnitude and taking the average of the two middle numbers. An exception is OHP as MOCAGE data is not available for this station so that the median refers to the middle number here. In addition, results from separate models are briefly discussed and shown in the main part of the manuscript to understand characteristics of the model ensemble output. However, it is beyond the scope of this study to describe the performance of each individual model in detail. The reader is referred to the Appendix for additional comparison Figures of individual model simulations and MAX-DOAS 20  data.

While the calculation of an ensemble median is a common approach to reduce individual model outliers, it is mainly used here for the sake of simplicity and presentation purposes, allowing easier overall evaluation of how the models compare to MAX-DOAS retrievals. The model ensemble is based on five of the seven models (though with partly different set-ups) which constitute the CAMS regional model ensemble (http://www.regional.atmosphere.copernicus.eu/) for which Marécal et 25  al. (2015) have shown that at least for ozone, the ensemble median performs on average best in terms of statistical indicators compared to the seven individual models and that the ensemble is also robust against reducing the ensemble size by one member. Statistical indicators for $NO_2$ (see Table 3 to 5) show that the ensemble median of the present study performs best in terms of overall correlation to individual MAX-DOAS measurements at each station. Compared to individual models for other statistical indicators and also comparisons for seasonal, diurnal and weekly cycles, reasonable results are achieved by the 30  ensemble median.

As the typical error on MAX-DOAS retrieved VCDs is around 20 %, but can be higher for sites with low trace gas concentrations like OHP or due to instrumental conditions (see Section 2.2), a conservative overall uncertainty of MAX-DOAS retrievals of 30 % is assumed for all stations within this manuscript and given along with the data plots, where appropriate.

Data products with more detailed uncertainty information are currently in development for example in the framework of the FRM4DOAS project (http://frm4doas.aeronomie.be/), and once available, this data and related uncertainty information should be used in future comparison studies.

## 4   Results

Figure 2 shows time series of AVK-weighted tropospheric $NO_2$ VCDs from MAX-DOAS and model ensemble data. The magnitude of VCDs from the measurements for Bremen and OHP is reproduced by the model ensemble. For Uccle and De Bilt, retrieved values tend to be larger than simulated ones. Low retrieved values appear overestimated at De Bilt and Bremen. At all of the four stations, measurements and simulations show large deviations for some of the time steps investigated. Larger $NO_2$ values inside individual pollution plumes are generally underestimated by the model ensemble, especially at Uccle and De Bilt. This is in agreement with Shaiganfar et al. (2015) who compared car MAX-DOAS measurements and OMI retrievals with a regional model (CHIMERE) and found that values inside emission plumes are systematically underestimated. The model ensemble may fail to reproduce these peaks due to errors in transport of $NO_2$ towards the stations or incomplete representation of atmospheric chemistry. An example of the latter would be overestimation of conversion to $HNO_3$, which may result in lower tropospheric $NO_2$ VCDs compared to MAX-DOAS if the transport is not happening quickly enough. Moreover, differences between simulations and retrievals may also arise from uncertainties of anthropogenic NOx emissions and horizontal resolution of model results (e.g. pollution sources may not be sufficiently resolved by the model simulations). Colette et al. (2014) compared regional model simulations with differing horizontal resolution and found that an increase in resolution leads to a better agreement with $NO_2$ in-situ data. However, as described in Section 1, MAX-DOAS observations are closer to regional model output in horizontal resolution than in-situ data. As expected, the magnitude of $NO_2$ VCDs is lowest at the rural station OHP, which is sometimes affected by near by pollution plumes that show up in the time series. Further investigation shows, that most of these peaks are associated with north-easterly wind directions and hence pollution sources to the north-east of the station such as the Po valley (Italy). At OHP, retrieved tropospheric $NO_2$ columns are generally a bit higher than simulated ones. At least for the summer period, this is in agreement with Huijnen et al. (2010) who showed that the GEMS regional model ensemble median underestimates background values of tropospheric $NO_2$ columns compared to OMI satellite retrievals. Note that we carried out a similar comparison to OMI for the model runs of the present study, which showed similar results as Huijnen et al. (2010) and is therefore not shown here (see Section 5).

The evolution of time series of tropospheric $NO_2$ VCDs is largely determined by the evolution of surface partial columns (see Figure 3) which already account for about 25 % of the magnitude of tropospheric $NO_2$ VCDs. In the present study, surface partial columns refer to the partial column of the lowest measurement layer (Bremen 50 m, De Bilt 180 m, Uccle 180 m, OHP 150 m above ground). As vertical profiles are not available from the MAX-DOAS output for De Bilt, comparisons of surface partial columns are not given for this station in the present manuscript. The same conclusions as for tropospheric $NO_2$ VCDs described in the previous paragraph arise for surface partial columns when comparing model ensemble to MAX-DOAS data. The negative bias found for OHP for trZZpospheric $NO_2$ VCDs is not present when looking at the surface partial column

time series for this station (see also Table 3 and Table 4 where most models are negatively biased at OHP for tropospheric columns but not for surface partial columns), indicating that $NO_2$ lifted above the ground level is underestimated compared to MAX-DOAS, pointing at uncertainties related to the transport of pollution and/or chemical conversion during transport.

Although there are larger differences between simulations and retrievals especially for individual pollution plumes, Figure 4 shows that frequency distributions of tropospheric $NO_2$ VCDs are similar for ensemble simulations and observations. However, for OHP the number of data values with tropospheric $NO_2$ VCDs lower than 1 x $10^{15}$ molec cm$^{-2}$ is significantly larger for model simulated values (about 1400 model values compared to about 200 observed data counts) in agreement with the negative bias in tropospheric columns described above.

Figure 5 shows model simulated and MAX-DOAS retrieved vertical profiles of $NO_2$ partial columns averaged over the whole time period of measurements together with a-priori profiles and AVKs for completeness. Averages of vertical profiles over different seasons are given in Figure A1, in order to investigate consistency between profiles throughout different times of the year. In general, differences between retrievals and simulations are largest for larger $NO_2$ partial columns, which means for the lower altitude layers and during the colder winter and autumn seasons. Many of the values simulated by individual models do not fall into the uncertainty range of MAX-DOAS retrievals assumed here. For example, SILAM largely overestimates $NO_2$ partial columns up to 1.5 km altitude at OHP, while MOCAGE (apart from the lowest observation layer) overestimates values up to about 1 km altitude at Uccle. Although model ensemble profiles show some differences to the retrievals regarding the shape and magnitude of the profiles, they also show the largest partial columns close to the surface for all of the three stations investigated. This result also shows up throughout different seasons.

As the sensitivity of MAX-DOAS retrievals is largest in the boundary layer, a feature which is independent of the retrieval method, we initially expected the application of column AVKs from the measurements to model simulations to be of crucial importance for evaluation results. However, further analysis showed that applying column AVKs to model $NO_2$ partial columns before summing these up in the vertical does not have a big impact on derived tropospheric $NO_2$ VCDs and therefore has a minor effect on the data analysis presented in this manuscript. Only AVK-weighted simulations of tropospheric $NO_2$ VCDs are therefore shown here. Statistical values (root mean squared error, bias, Pearson correlation coefficient) which will be described below are quite similar for AVK weighted model ensemble VCDs and those from non AVK-weighted ones. One of the reasons for this is that (as shown by Figure 5 and Figure A1), AVKs are close to 1 around the boundary layer where MAX-DOAS instruments have the highest sensitivity (generally a bit larger than one close to the surface and smaller than one higher up which has a balancing effect) and that the vertical shape of the column AVK curve is in principal agreement with the shape of simulated $NO_2$ partial columns. At altitudes above roughly 1 km, AVKs are on average for some stations significantly smaller than one, but simulated $NO_2$ partial columns are also significantly smaller at these altitudes compared to lower levels, so that the contribution to the tropospheric column is limited. At higher altitudes, MAX-DOAS retrievals tend to follow the a-priori, while retrievals in the boundary layer are not much influenced by the a-priori in general. This is in contrast to the situation for satellite observations of tropospheric $NO_2$, which usually have a minimum of the AVK in the boundary layer, i.e. where the largest fraction of $NO_2$ is usually located in polluted situations. A-priori profiles used within the MAX-DOAS retrievals (see

Section 2.2) are in principal agreement with the ones simulated by the models. The vertical weighting caused by application of AVKs to partial columns does therefore not significantly impact on derived tropospheric $NO_2$ VCDs.

Scatter density plots of tropospheric $NO_2$ VCDs from MAX-DOAS against model values corresponding to the time series displayed by Figure 2 are shown in Figure 6 (see Figure A2 for individual model results). Statistical values (root mean squared error, bias, Pearson correlation) and least squares regression lines are given along with the plots to draw further conclusions on the ability of the model ensemble to reproduce MAX-DOAS retrievals and are listed in Table 3 (together with statistics on ensemble members). Statistical values for surface partial columns are given in Table 4. The ensemble median performs best in terms of overall correlation with values between 45 to 75 % (compared to 35 to70 % for individual models) for tropospheric $NO_2$ VCDs for all stations, the highest correlation is found for Uccle. Note however, that for other statistical indicators, some of the individual models perform better. Correlations are generally lower than the ones based on tropospheric columns for surface partial columns which are on the order of 40 % for Bremen and OHP, but much higher again for Uccle ($\sim$60 %) for the ensemble. As expected from the comparisons described above, the model ensemble has a negative bias of about -0.3 and -2 x $10^{15}$ molec cm$^{-2}$ for OHP and Uccle, respectively, and a positive bias of about 1 x $10^{15}$ molec cm$^{-2}$ for De Bilt and Bremen for tropospheric columns. The largest rms and bias (10.5 and 5 x $10^{15}$ molec cm$^{-2}$, respectively) are found for LOTOS-EUROS at De Bilt. Considering that values for OHP are generally smaller than for the three urban sides, SILAM also shows a considerably high rms and bias (2.6 and 1.2 x $10^{15}$ molec cm$^{-2}$, respectively) at this station. Vertical profile comparisons described above show that the overestimation mainly occurs at altitudes up to about 1.5 km. Our findings agree with Vira and Sofiev (2015) who found that SILAM tends to overestimate $NO_2$ at rural sites based on in-situ data and concluded that this is due to an overestimation of the lifetime of $NO_2$, which is also consistent with findings by Huijnen et al. (2010). For surface partial columns, biases are negligibly small for OHP and Bremen for the ensemble and most of the individual models, while the ensemble is negatively biased by about 1 x $10^{15}$ molec cm$^{-2}$ at Uccle. The largest rms and bias in surface partial columns are found for EMEP at Uccle (3.3 and -1.8 x $10^{15}$ molec cm$^{-2}$, respectively). The spread between models and observations is large for some individual data points. Regression lines show that the model ensemble tends to overestimate low and underestimate high tropospheric $NO_2$ VCDs. The underestimation of larger tropospheric $NO_2$ VCDs is most pronounced for De Bilt, followed by Uccle.

Figure 7 shows comparisons between MAX-DOAS and the model ensemble of wind directional distributions of average tropospheric $NO_2$ VCDs based on wind measurements from station data (note that further analysis has shown a good agreement between measured wind speeds and wind directions and those of the simulations). Changes of $NO_2$ mean values from one wind direction bin to another are reproduced well by the model ensemble (and in general also by ensemble members, see Figure A3), with an overall slightly better agreement with retrievals for tropospheric $NO_2$ VCDs compared to surface partial columns (not shown). Both, MAX-DOAS and model ensemble show the highest $NO_2$ mean values for wind directions mainly where influence from pollution sources is expected (i.e. Ruhr area to the south-east of De Bilt, the Bremen city centre to the south-west of the Bremen MAX-DOAS, Brussels city centre to the north-east of Uccle, the Po valley to the north-east of OHP, see Section 2.2). As for the time series comparisons described above, differences between observations and model results could be related to model uncertainties in simulating transport of pollution towards the measurement stations and chemistry.

Uncertainties in anthropogenic emissions and background $NO_2$ VCDs may add up to differences between models and MAX-DOAS for wind directional distributions.

Comparisons for seasonal cycles (i.e. monthly averages) of tropospheric $NO_2$ VCDs are given in Figure 8 together with corresponding statistical values in Table 5. The number of MAX-DOAS measurements available for each month is given at the top y-axis of each seasonal cycle plot as an indicator of statistical significance. The number of data values is also shown for diurnal and weekly cycle Figures which will be discussed below. There is a good agreement between MAX-DOAS and the model ensemble for Uccle regarding the magnitude of $NO_2$ VCDs and seasonality, with simulated ensemble median values within the estimated uncertainty interval of the retrievals. The same is true for De Bilt, apart from the strong overestimation of MAX-DOAS retrieved values for January, March and April. The latter may be explained by the low number of observations available during these compared to other months. The model ensemble overestimates seasonal cycles for Bremen and OHP. More explicitly, there is an overestimation of wintertime values while summertime values are better reproduced by the model ensemble. This may indicate that the model ensemble overestimates production of OH via photolysis of $O_3$ when less light is available, as OH acts as a sink for $NO_2$. The latter may also result from errors in simulating clouds and related photochemistry during the colder season. It may also point to an overestimation of anthropogenic emissions or inappropriate scalings of these. The former would be in agreement with Petetin et al. (2015), who found that anthropogenic NOx emissions from the TNO emission inventory (on which MACC emissions are based on) are overestimated, but those results apply to the Paris region only. Huijnen et al. (2010) compared an ensemble of regional and global models to satellite data over Europe and found an overestimation of seasonal cycles by the simulations, which is in agreement with results for Bremen and OHP shown in the present manuscript. However, according to Huijnen et al. (2010) model values were closer to satellite retrievals during winter, whereas for summer a strong underestimation was found, while comparisons to Dutch surface observations showed that this could be partly attributed to a high bias of satellite retrievals in summer at least over the Netherlands. In the present study, the spread between individual models is quite large for OHP indicating that some of the models perform better than others. Looking at the spread between individual models also shows that seasonal cycles are generally more pronounced compared to the other model runs and retrievals for LOTOS-EUROS and MOCAGE. Especially LOTOS-EUROS largely overestimates the observed seasonal cycle at OHP. Low to moderate correlations in seasonal cycles are found for De Bilt, followed by moderate ones for Bremen. All models perform well in terms of correlation at Uccle and OHP (values around 0.8).

Figure 9 shows comparisons of diurnal cycles for the whole time series. Overall, the model ensemble fails to reproduce diurnal cycles for all stations, reflected by generally low correlations (Table 5) for all models at De Bilt, Bremen and OHP. All models show negative correlations at De Bilt, while some of the models only reach negative correlations at Bremen as well. MAX-DOAS retrieved values increase from the morning towards the afternoon, while simulated values in general decrease from the morning towards the afternoon. At Uccle however, high or at least moderate correlations are achieved. CHIMERE performs best in terms of correlation at Uccle and OHP (0.92 and 0.6, respectively). For this model, diurnal scaling factors of traffic emissions have been developed by analyzing measurements of $NO_2$ in European countries (Menut et al., 2013; Marécal et al., 2015). Although most of the model values fall within the estimated uncertainty interval of MAX-DOAS retrievals, the shape of diurnal cycles differs between observations and simulations. The ensemble shows a strong peak during the morning rush hour

around 8 am for Bremen, which is not confirmed by MAX-DOAS retrievals. In contrast to this, measurements show a maximum around 2 pm in the afternoon which coincides with a very weak local maximum simulated by the model ensemble. Looking at diurnal cycles for different seasons shown in Figure A4 and A5 reveals that these are in general much better reproduced for spring and summer compared to autumn and winter for all stations. This is in agreement with results for seasonal cycles

described in the previous paragraph. Weak morning rush hour peaks are also simulated for the rural station OHP, which is not in agreement with the measurements. The morning rush hour peaks for Bremen and OHP occur for all models with the exception of SILAM for OHP, which however strongly overestimates values (by a factor of 1.5 to 2 for diurnal cycle values averaged over the whole time series) for this station, resulting in a bias of $1.3 \times 10^{15}$ molec cm$^{-2}$ (see Table 5). The peak at 8 am for Bremen is most pronounced for EMEP-MACCEVA, MOCAGE and LOTOS-EUROS. Individual model runs show the same shape of

the diurnal cycle for Bremen, while the shape of diurnal cycles differs for OHP. Moreover, large differences regarding the magnitude of simulated values occur for both stations. As described in Section 2.1, all models use the same emission inventory as a basis, except the EMEP run. There is a strong difference between the magnitude of the values simulated by EMEP and EMEP-MACCEVA specifically for the diurnal cycle at Bremen (while the shape of the cycles is similar), which could be either related to the difference in resolution or different emission inventories incorporated in both of the two runs. The differences in

diurnal cycles between model simulations and retrievals as well as between individual model runs could mean that the different scalings of NOx emissions applied by each model to account for diurnal variations are not appropriate, maybe in combination with uncertainties in vertical scalings. This should be investigated in future modelling studies. For example, according to Mailler et al. (2013) improving efficient emission heights is a key factor for improving background atmospheric composition simulated by chemistry transport models. However, the disagreement between simulated and measured values as well as the

disagreement between individual model runs may also point to problems regarding photochemistry and treatment of boundary layer mixing. Differences in transport of pollution towards the stations during the morning and evening may add up to model uncertainties, especially for the rural station OHP where different shapes of diurnal cycles for individual model runs may also result from pollution transported from urban surrounding areas towards the station. Comparing to surface station measurements of ozone, Marécal et al. (2015) found that statistical indicators of model performance for MACC-II regional models show a

pronounced diurnal cycle (best performances at 15 UTC, worst ones at 18 UTC) and attributed this to uncertainties in the diurnal cycle of ozone precursor emissions.

Figure 10 shows comparisons of diurnal cycles for weekends (Saturdays and Sundays) only. A Figure of diurnal cycles for weekdays only shows very similar results as Figure 9 (and is therefore not shown here), meaning that overall diurnal cycles are mainly driven by weekday emissions. At the three urban stations, MAX-DOAS retrieved diurnal cycles show a different shape

for weekends compared to diurnal cycles for the whole week (and hence weekdays only). This is in contrast to model simulated diurnal cycles, which do not change much going from cycles for the whole week to cycles for weekends only, apart from a general decrease in values towards weekends for both retrieved and simulated tropospheric NO$_2$ VCDs. As expected, MAX-DOAS retrieved diurnal cycles are rather flat for weekends only at the urban stations, as emissions from traffic and industry are reduced during weekends compared to weekdays (e.g. Elkus et al., 1977; Beirle et al., 2003; Ialongo et al., 2016). As

the shape of simulated diurnal cycles is similar for weekdays versus weekend, the difference between retrieved and simulated

trends in tropospheric columns from morning to afternoon hours is reduced for weekends only resulting in significantly higher and positive correlations for diurnal cycles during weekends compared to weekdays for the ensemble at these stations (see Table 5). At Uccle, correlations are equally high (about 80 %) for weekdays and weekends, which is due to the fact that the shape of retrieved diurnal cycles is also similar. Correlations are also significantly higher for the background station OHP for weekends for the ensemble, mainly due to a better agreement in the development from the afternoon towards the evening during weekends. However, visually/by eye, the agreement between simulations and retrievals is similar for weekdays and weekends for this station. The results described above show that models fail to reproduce observed changes in diurnal cycles towards the weekend at urban stations, indicating that different diurnal scalings should be applied to emissions for weekdays and weekends. It should be tested in future simulations if switching off diurnal scalings during weekends leads to an improvement in model performance compared to MAX-DOAS.

Weekly cycle comparisons are presented in Figure 11 (see Figures A6 and A7 for different seasons). In contrast to diurnal cycles, weekly cycles and their seasonal variation measured by MAX-DOAS are much better simulated, reflected by high correlations (Table 5) for the ensemble at all stations. Both, MAX-DOAS and the model ensemble, show a decrease in tropospheric $NO_2$ VCDs towards the weekend when there is less traffic especially for the urban stations De Bilt, Bremen and Uccle. However, this observed weekly cycle is stronger than the simulated one, a feature which is most pronounced for Bremen. This is in agreement with Vlemmix et al. (2015) who also found an underestimation of the weekly cycle when comparing LOTOS-EUROS simulations to MAX-DOAS retrievals for De Bilt. As expected, only a very weak weekly cycle is observed by MAX-DOAS and simulated by the models for the rural station OHP. Note that maxima of weekly cycles for specific days may just be coincidence due to data sampling times. Beirle et al. (2003) investigated weekly cycles of tropospheric $NO_2$ based on GOME satellite observations and found a decrease in values of up to about 50 % towards Sundays over polluted regions and cities in Europe. This is in principal agreement with results of the present study, although the choice of the cities is different.

Comparing Table 3 and 5 shows, that the overall correlations reached at all stations are mainly driven by seasonal and weekly cycles, while significantly lower and in many cases negative correlations are found for diurnal cycles which decreases overall correlations. An exception for the latter is Uccle, where good correlations are also found for diurnal cycles.

## 5  Summary and conclusions

In this study, comparisons between $NO_2$ columns simulated by five regional models and retrieved from MAX-DOAS measurements for four European MAX-DOAS stations have been presented. The reasons for differences between model results and observations found by the comparisons are discussed here in a general sense and need to be further investigated by carrying out additional dedicated model runs in future modelling studies. In general, differences between simulated and retrieved tropospheric $NO_2$ VCDs as well as surface partial columns found in this study could result from model uncertainties in chemistry and meteorology or a combination of both. Moreover, errors related to NOx emission inventories or uncertainties in tropospheric MAX-DOAS retrievals may also contribute to differences between simulated and retrieved values found in this study.

Our analysis shows that in general and on average the model ensemble does well represent tropospheric NO$_2$ amounts observed by MAX-DOAS. However, many points to evaluate arise from the MAX-DOAS based comparisons. Tracking down the reasons for differences between simulations and retrievals and adjusting model runs accordingly (in case of differences caused by errors in simulations rather than uncertainties of the retrievals) could improve model performance substantially.

Moderate correlations around 60 % are found for tropospheric NO$_2$ VCDs at each station for the ensemble. Time series comparisons and corresponding scatterplots show that uncertainties in simulating pollution transport towards the stations is a likely reason for the underestimation of MAX-DOAS retrieved pollution peaks by the model ensemble. This may also lead to the weak simulated morning rush hour peak for the rural station OHP, which is not confirmed by the retrievals. In fact, for OHP, a diurnal cycle representative of a remote background NO$_2$ station would be expected. However, comparisons of wind

directional distributions of tropospheric NO$_2$ VCDs and surface partial columns show a good agreement between simulations and measurements. This indicates that transport of pollution towards the stations is, on average, well represented by the models.

Comparisons of vertical profiles show that the main source of the scatter between measurements and simulations is not due to incorrect representation of the vertical NO$_2$ distribution. Hence, there are no large differences between comparisons which do not make use of column AVKs for calculating model VCDs and those based on more accurate column AVK-weighted values.

The latter result was not expected as the sensitivity of the MAX-DOAS profile retrievals is much larger close to the surface than at altitudes larger than approximately 1 km.

Seasonal cycles are overestimated by the model ensemble. Simulation uncertainties in photochemistry and/or in monthly scalings of emissions are conceivable explanations for this. As MAX-DOAS measurements are carried out throughout the whole course of a day during daylight and are hence available with a comparatively high resolution in time, it is (in contrast

to many other approaches) possible to compare diurnal cycles derived from simulations and measurements. This reveals that models fail to reproduce the shape of diurnal cycles for all stations as well as the observed change in diurnal cycles from weekdays towards weekends at urban stations, which most likely points to uncertainties in diurnal scalings of emissions. Improving model results for diurnal cycles could potentially have a strong impact on all other comparisons shown in this manuscript and hence may further improve model performance. This is in agreement with Mues et al. (2014) who found an

improvement of correlations between LOTOS-EUROS and in-situ data when applying a time profile to emissions. It should be tested in future studies if switching off diurnal scalings during weekends leads to an improvement in model performance compared to MAX-DOAS. The largest differences to MAX-DOAS retrieved seasonal and diurnal cycles generally occurred for LOTOS-EUROS and MOCAGE at Bremen and De Bilt and also for EMEP-MACCEVA at Bremen. LOTOS-EUROS and SILAM showed the largest differences to retrieved diurnal and seasonal cycles for the background station OHP. However,

weekly cycles are better represented by the model ensemble, which indicates that applied scalings of emissions on a daily basis are at least more appropriate than hourly ones. However, the models generally underestimate the decrease in tropospheric NO$_2$ VCDs towards the weekend. This decrease was reproduced much better by SILAM compared to the other models. The comparisons to MAX-DOAS also showed that this model overestimates values at the background station OHP, in agreement with a study by Vira and Sofiev (2015) who related this to an overestimation of the lifetime of NO$_2$.

In addition to the MAX-DOAS comparisons shown in the present study, we also carried out a comparison between the regional models and OMI (Levelt et al., 2006) satellite retrievals looking at maps of monthly means for a winter and summer month (February and August 2011, respectively) falling into the time period investigated by the present study. We found similar results as Huijnen et al. (2010) which are therefore not shown here, i.e. an underestimation of tropospheric $NO_2$ columns over background regions during summer (in agreement with the general underestimation of means over summer months compared to MAX-DOAS shown by seasonal cycles for OHP for all models except SILAM) and a generally better agreement between satellite retrievals and models over pollution hotspots around Benelux countries, an underestimation however of values over large parts of Germany and over the Po valley in many of the model runs. Some of the models also overestimated values to the south and south-east of OHP (roughly between Marseille and Genua along the southern coast of France) compared to OMI. However, due to the generally short lifetime of $NO_2$, to properly relate uncertainties in the simulations over emission hotspots indicated by the OMI based comparisons to the ones derived from MAX-DOAS based comparisons would generally require investigating transport patterns of individual model runs with much higher time resolution around the MAX-DOAS sites, which is not provided by the satellite data (only one OMI orbit per day over the stations).

Our evaluation demonstrates that the large number of measurements available from the current MAX-DOAS network constitutes a useful data source for investigating the performance of regional models. In contrast to other measurements usually applied for evaluation of regional models, MAX-DOAS data are available with comparatively high resolution in time. Furthermore, MAX-DOAS retrievals are representative of a larger volume of air and are therefore much better suited for regional model evaluation than in-situ data.

The horizontal grid spacing (Table 1) differs for the 6 model runs evaluated in the present study, with a resolution of approximately 9x7 km$^2$ for the highest resolution run (LOTOS-EUROS) and 50x50 km$^2$ for the coarsest one (EMEP). The resolution of the remaining model runs is approximately 20x20 km$^2$. As described in Section 2.2, the horizontal averaging volume of MAX-DOAS retrievals strongly depends on aerosol loading, viewing direction and wavelength (Richter et al., 2013). As a rough estimate, it ranges from 5 to 10 km for the stations used in the present study. Therefore, the horizontal averaging volume is (apart from the coarsest resolution run) expected to be either on the same spatial scale as the horizontal model resolution or by a factor of 1 to 4 smaller. From the latter (i.e. horizontal averaging volume of MAX-DOAS smaller than model resolution) one would expect an underestimation of enhancements in tropospheric columns observed by MAX-DOAS in case of horizontal changes in tropospheric $NO_2$ columns below the model resolution and, similarly, an overestimation of local minima in tropospheric $NO_2$ columns. However, in reality, the comparison between horizontal averaging volume of MAX-DOAS and horizontal resolution of the models is much more complicated, as MAX-DOAS instruments usually measure in one azimuthal pointing direction meaning that measurements are performed only on a specific line of sight whereas model simulations are performed for three dimensional grid boxes. This could for example mean that a pollution plume with a horizontal extent on the order of the model resolution and hence showing up in the simulations is missed by the line of sight of the MAX-DOAS instrument. It would therefore be desirable to perform multiple MAX-DOAS measurements over a range of different azimuthal angles for each station and use these in future model to MAX-DOAS comparison studies.

A pollution plume and related increase in the time series of tropospheric $NO_2$ VCDs observed by MAX-DOAS would be expected to be reproduced better by model runs with higher horizontal resolution compared to lower resolution runs. The lifetime of $NO_2$ is also expected to increase with model resolution. However, in the present study, the LOTOS-EUROS run with significantly higher horizontal resolution than the other runs in general did not perform better than lower resolution runs which can probably be explained by its low number of vertical layers. Similarly, the EMEP run with significantly lower horizontal resolution did not perform worse than higher resolution runs, which shows that other differences between the models such as chemistry schemes and treatment of emissions strongly impact on comparison results. It would be interesting to investigate the ability of the models to predict the scales of $NO_2$ spatial variations derived from time scales of $NO_2$ variations and wind speeds in the context of model resolution in a future study. Moreover, one could investigate the ability of the models to distribute $NO_2$ in the vertical in terms of characteristic layer height of $NO_2$, which is (in addition to other factors like vertical distribution of emissions or boundary layer schemes) expected to be affected by vertical resolution of the models.

Comparison results of this study could be compared and complemented by further data sources where possible. Future investigations of regional model performance may also include application of stricter quality filters on the MAX-DOAS data to reduce the impact of retrieval uncertainty. As the discussion here is based on results of five regional models used within CAMS for four European stations, similar comparisons to other regional models or other model set-ups as well as for more MAX-DOAS sites should follow. As the stations investigated in the present study have, apart from the rural background station OHP, rather similar meteorological and pollution conditions, investigation of stations over a broader range of different conditions would be desirable. Further comparison studies could for instance include stations at pollution hotspots in the Mediterranean such as Athens with strong smog conditions especially during summer and clean mountain sites. The impact of different model set-ups and different anthropogenic emission inventories on comparison results should be tested in order to improve model performance. Moreover, the complex influence of clouds on comparison results could be investigated.

To track down reasons for the reported uncertainties of regional model simulations constitutes the main challenge for future studies. This could be achieved by running models with different chemistry schemes combined with different resolutions where possible (uncertainties in chemistry such as lifetime of $NO_2$), running models with and without scaling of emissions in time and for specific seasons or days only (uncertainties in seasonal, diurnal and weekly cycles related to emissions), performing runs with varying vertical scalings of emissions (uncertainties in injection heights) and carrying out runs with varying boundary layer physics (uncertainties of $NO_2$ profiles due to mixing of emissions in the boundary layer and transport therein). Especially LOTOS-EUROS and MOCAGE showed large differences to the MAX-DOAS retrieved seasonal and diurnal cycles for Bremen and De Bilt and also EMEP-MACCEVA for Bremen, so that the impact of different set-ups in emissions and chemistry is expected to be more pronounced compared to the other models at these stations.

## 6 Code availability

Source code and test data sets for the Open Source EMEP/MSC-W model are available at https://github.com/metno/emep-ctm or by contacting EMEP/MSC-W (emep.mscw@met.no). The SILAM code is available on request from the authors

(mikhail.sofiev@fmi.fi, julius.vira@fmi.fi). The MOCAGE results in the present paper are based on source code which is presently incorporated in the MOCAGE model. The MOCAGE source code is the property of Météo-France and CERFACS, and it is based on libraries that belong to some other holders. The MOCAGE model is not open source and routines from MOCAGE cannot be freely distributed. CHIMERE is an open source code protected under the GNU General Public license.

It can be found at http://www.lmd.polytechnique.fr/chimere/. LOTOS-EUROS is downloadable free of charge after signing a license agreement. All information concerning the LOTOS-EUROS code is available on the website (http://lotos-euros.nl), for further information the reader can contact Dr. A. Manders (astrid.manders@tno.nl).

*Acknowledgements.* This study was funded by the European Commission under the EU Seventh Research Framework Programme (grant agreement no. 283576, MACC II), the EU Horizon 2020 Research and Innovation programme (grant agreement no. 633080, MACC-III)

and the Copernicus Atmosphere Monitoring Service (CAMS), implemented by the European Centre for Medium-Range Weather Forecasts (ECMWF) on behalf of the European Commission. It was also funded in part by the University of Bremen. The LOTOS-EUROS work was carried out within the ESA project, GLOB-EMISSION (grant number AO/1-6721/11/I-NB). BIRA-IASB MAX-DOAS observations at Uccle and OHP were financially supported by the projects AGACC-II (BELSPO, Brussels) and NORS (EU FP7; contract 284421). We thank the German Weather Service/ Deutscher Wetterdienst for providing wind in-situ data for Bremen through their website at http://www.dwd.de.

We are also gratefull to people behind the wind in-situ data at Uccle and OHP for providing these measurements through the webpages http://uvindex.aeronomie.be and http://pc-meteo.obs-hp.fr/intervalle.php, respectively.

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

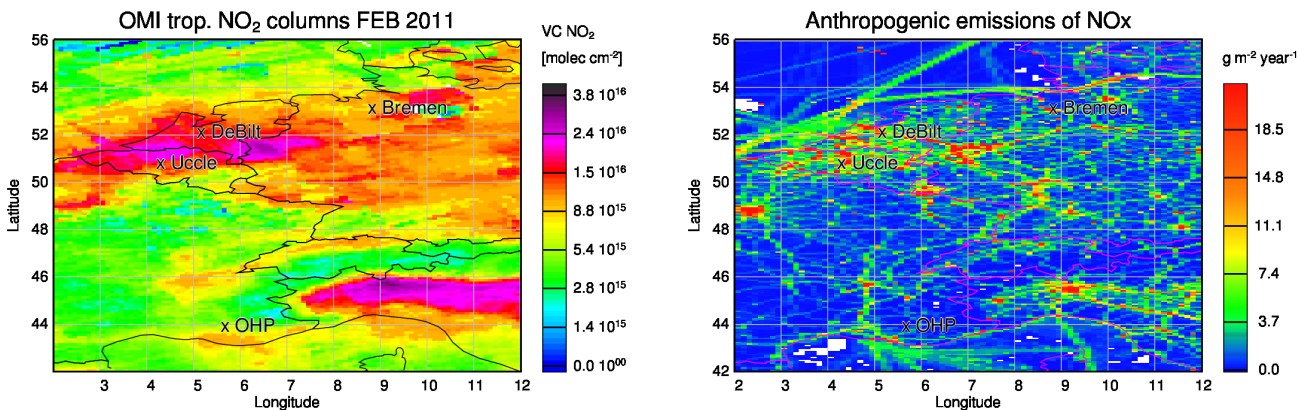

**Figure 1.** Maps of (left) average tropospheric $NO_2$ VCDs [molec $cm^{-2}$] observed by OMI for February 2011 and (right) TNO/MACC-II anthropogenic NOx emissions [g $m^{-2}$ $year^{-1}$] over Europe. Location of MAX-DOAS measurement sites investigated in this study are marked by black crosses on the maps. The satellite data has been gridded to 0.1° lat x 0.1° lon, the resolution of the emission database is 0.125° lat x 0.0625° lon.

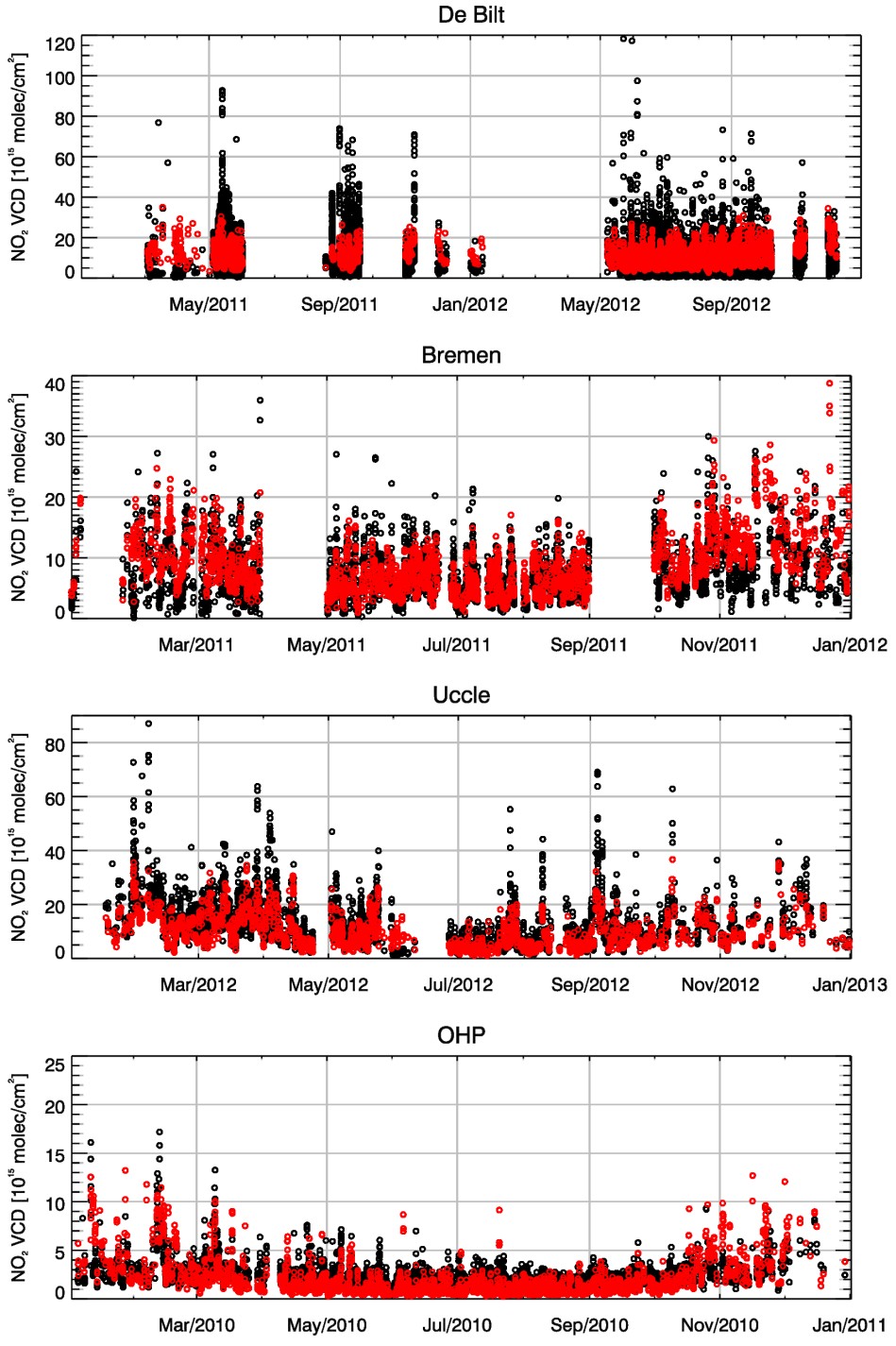

**Figure 2.** Time series of AVK-weighted tropospheric $NO_2$ VCDs [$10^{15}$ molec cm$^{-2}$] from (black circles) MAX-DOAS and (colored circles) model ensemble hourly data for (from top to bottom) De Bilt, Bremen, Uccle and OHP.

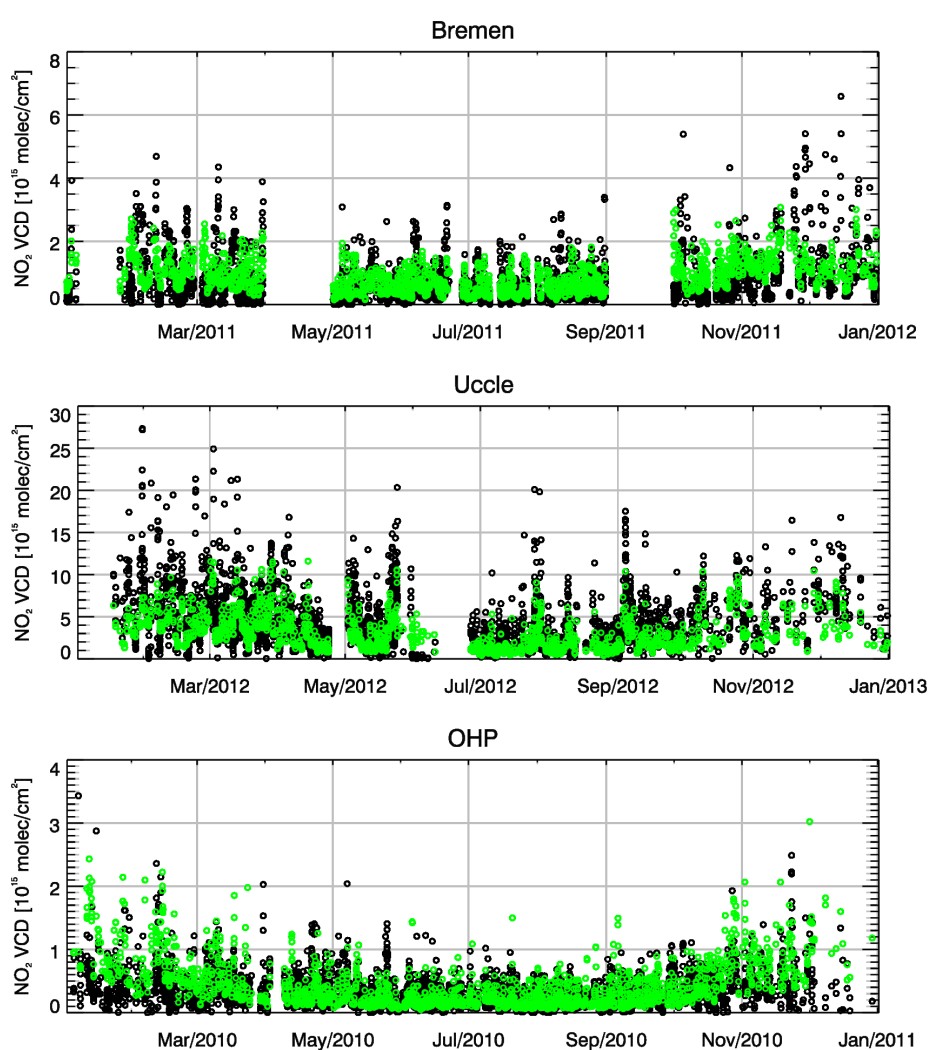

**Figure 3.** As in Figure 2 but for $NO_2$ surface partial columns [$10^{15}$ molec cm$^{-2}$]. Surface partial columns from MAX-DOAS are not available for De Bilt for the investigated time period.

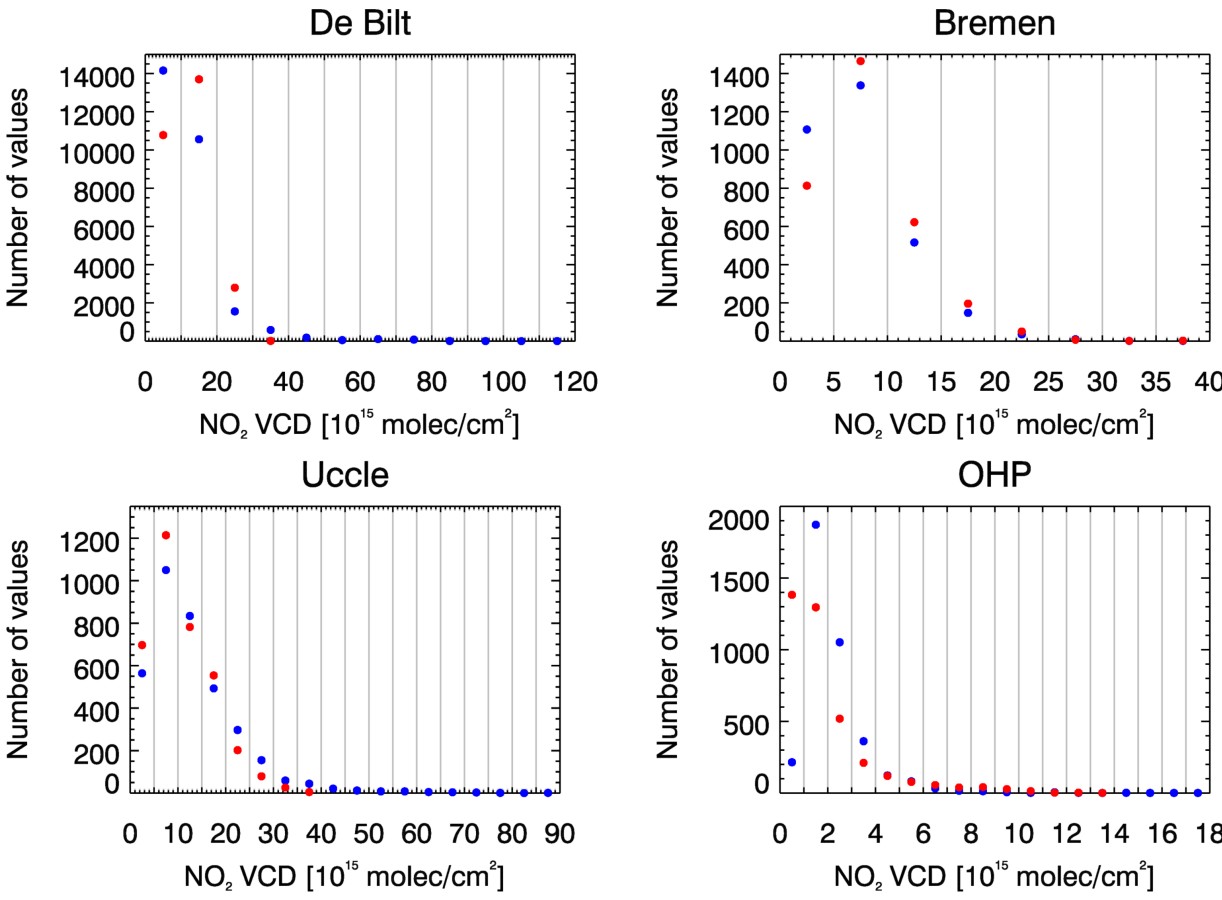

**Figure 4.** Frequency distributions of AVK-weighted tropospheric $NO_2$ VCDs [$10^{15}$ molec cm$^{-2}$] from (blue) MAX-DOAS and (red) model ensemble data for (top left) De Bilt, (top right) Bremen, (lower left) Uccle and (lower right) OHP. The distance between vertical grey lines on the x-axis corresponds to the size of the bins used to calculate the number of values given on the y-axis.

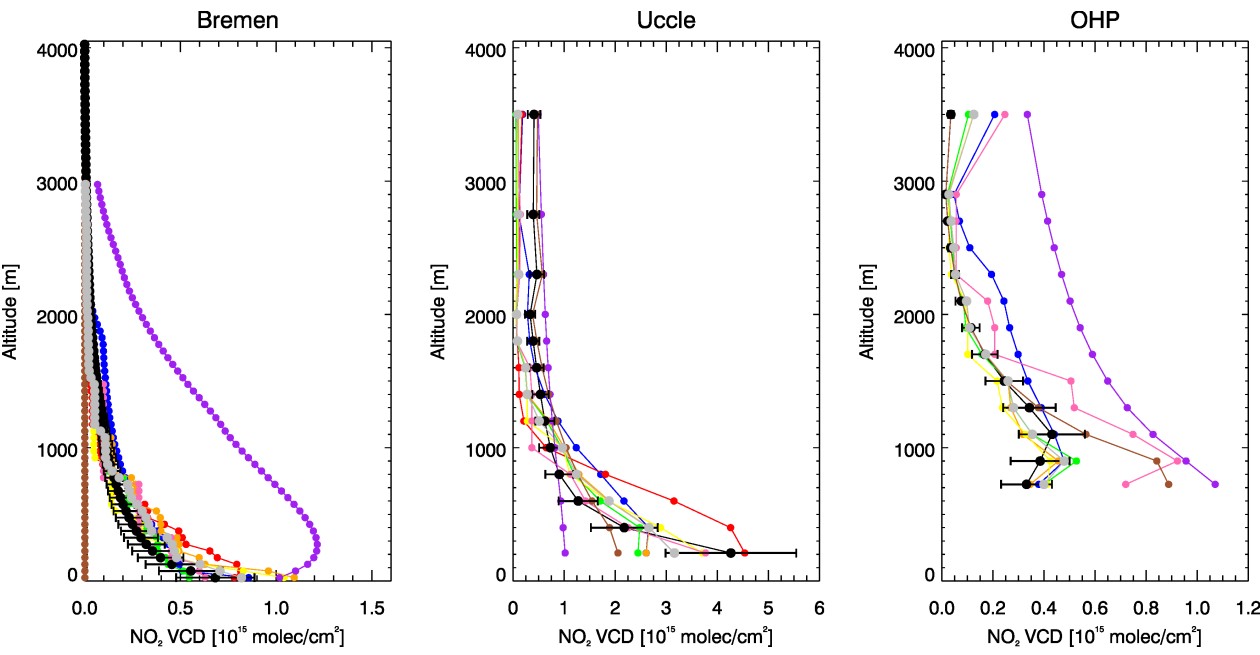

**Figure 5.** Average vertical profiles of NO$_2$ partial columns [$10^{15}$ molec cm$^{-2}$] from (black) MAX-DOAS, (brown) a priori used for MAX-DOAS retrievals, (gray) model ensemble median, (blue) LOTOS-EUROS, (yellow) CHIMERE, (green) EMEP, (orange) EMEP-MACCEVA, (pink) SILAM and (red) MOCAGE as well as (purple) column averaging kernels [unitless] for (left) Bremen, (middle) Uccle and (right) OHP. Black error bars refer to the uncertainty associated with the MAX-DOAS retrievals (assumed to be 30 % for all stations). MAX-DOAS vertical profiles are not available for De Bilt for the investigated time period.

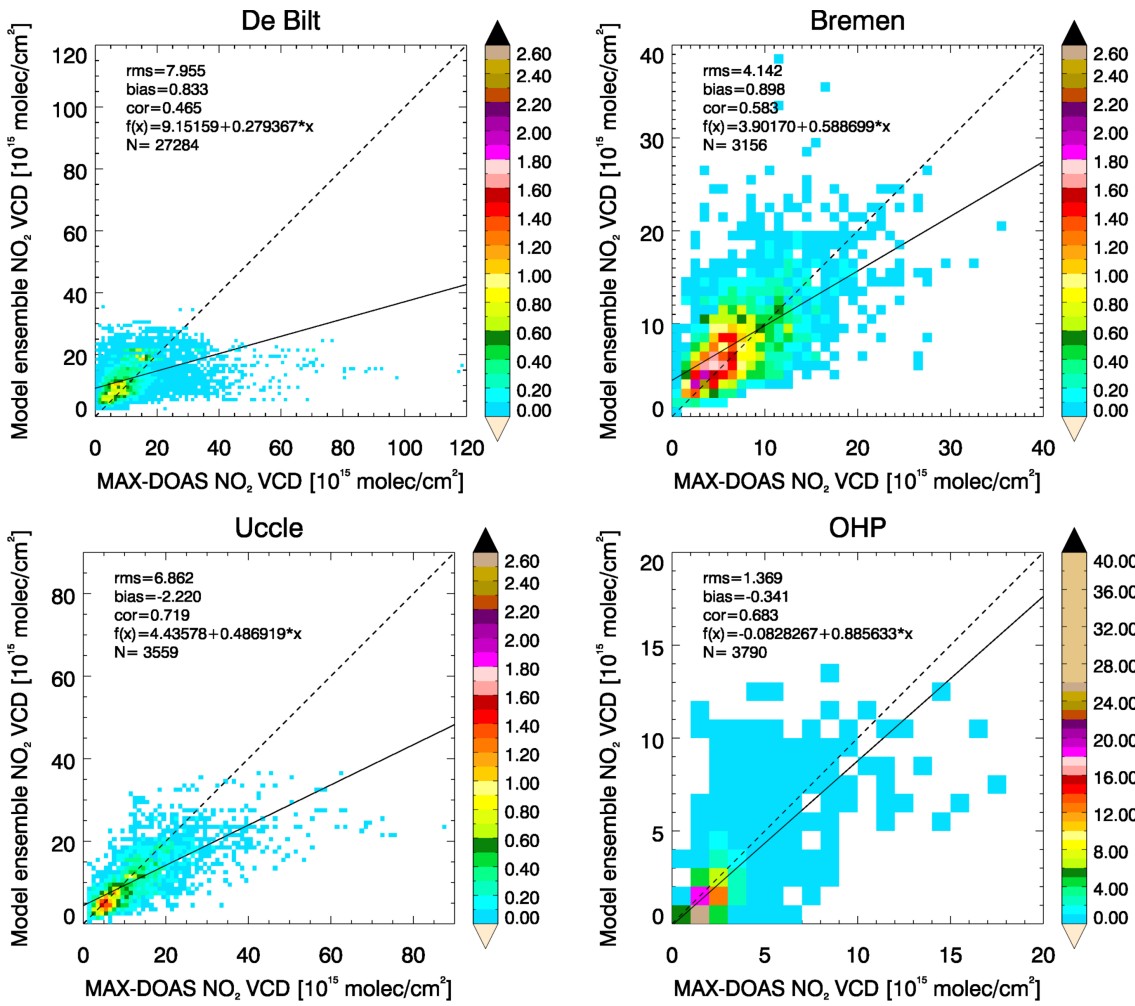

**Figure 6.** Scatter density plots of AVK-weighted tropospheric $NO_2$ VCDs [$10^{15}$ molec cm$^{-2}$] from MAX-DOAS against model ensemble data for (top left) De Bilt, (top right) Bremen, (lower left) Uccle and (lower right) OHP. The data is shown for different bins with a size of $10^{15}$ molec cm$^{-2}$ and is colored according to the number of data points per bin [%]. The dashed line is the reference line (f(x)=x). The solid line is the regression line (see top left of each plot for f(x) of this line). The root mean squared error (rms) [$10^{15}$ molec cm$^{-2}$], bias [$10^{15}$ molec cm$^{-2}$], Pearson correlation coefficient (cor, not squared) as well as the number of data points N are given at the top left of each plot.

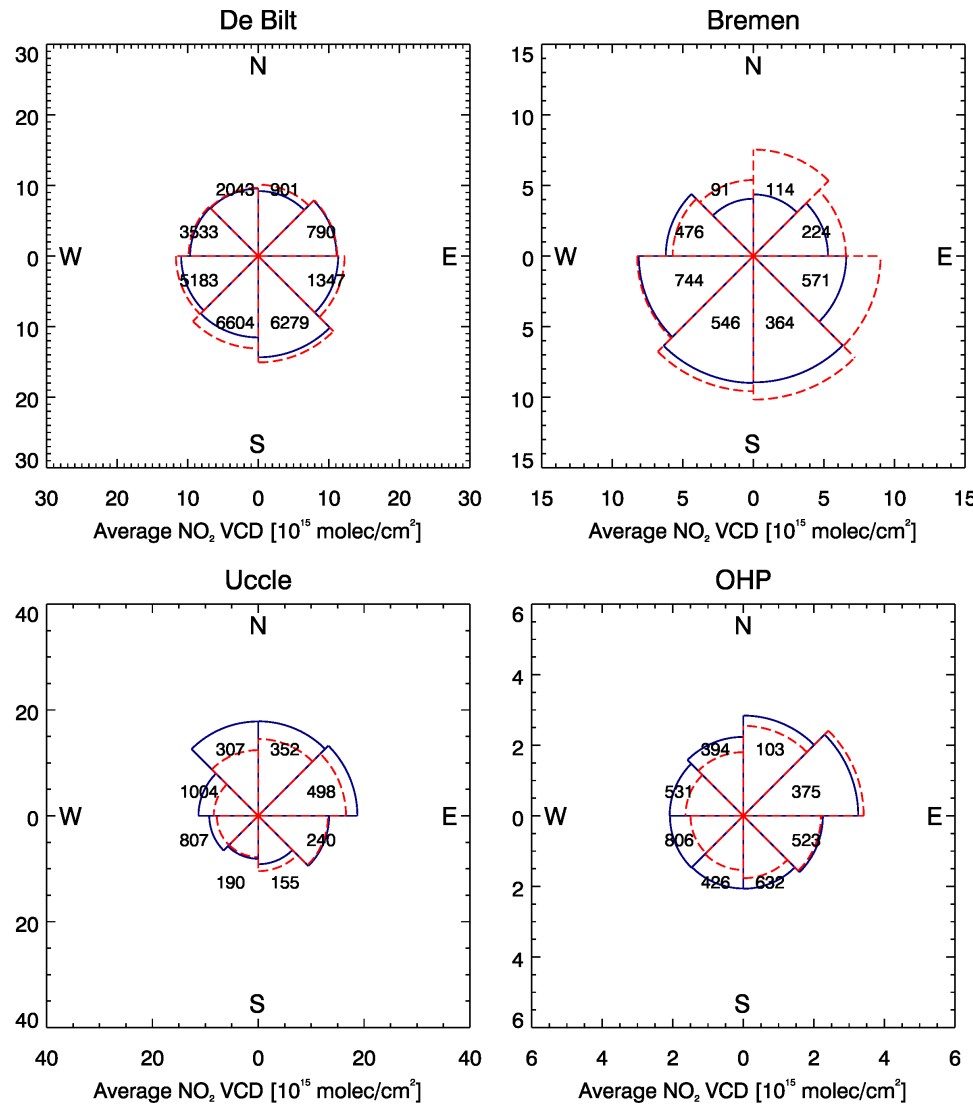

**Figure 7.** (a) Average AVK-weighted tropospheric $NO_2$ VCDs [$10^{15}$ molec cm$^{-2}$] in 45° wide wind direction bins from (blue solid lines) MAX-DOAS and (red dashed lines) model ensemble data for (top left) De Bilt, (top right) Bremen, (lower left) Uccle and (lower right) OHP. Wind directions correspond to the direction towards the station and are taken from weather station measurements. The numbers close to the centre of each plot refer to the number of data values used for calculating average values for each bin.

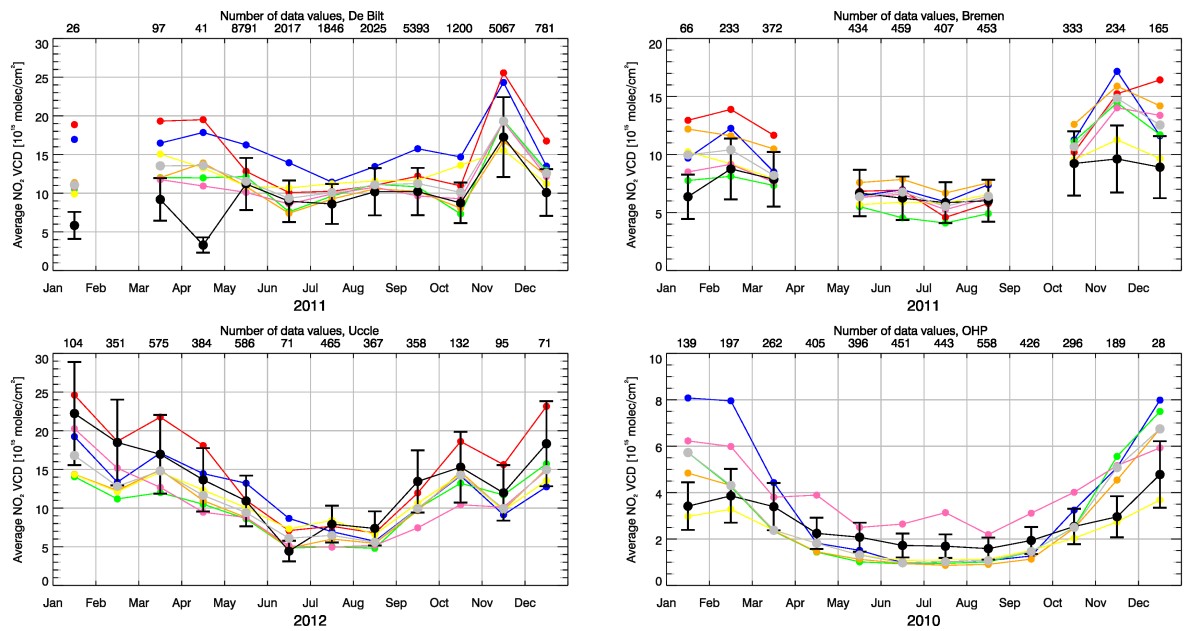

**Figure 8.** Seasonal cycles (monthly averages) of AVK-weighted tropospheric $NO_2$ VCDs [$10^{15}$ molec cm$^{-2}$] from (black) MAX-DOAS, (gray) model ensemble median, (blue) LOTOS-EUROS, (yellow) CHIMERE, (green) EMEP, (orange) EMEP-MACCEVA, (pink) SILAM and (red) MOCAGE for (top left) De Bilt, (top right) Bremen, (lower left) Uccle and (lower right) OHP. Black error bars refer to the uncertainty associated with the MAX-DOAS retrievals (assumed to be 30 % for all stations). The number of data values used for calculating average values is shown at the upper x-axis of each plot.

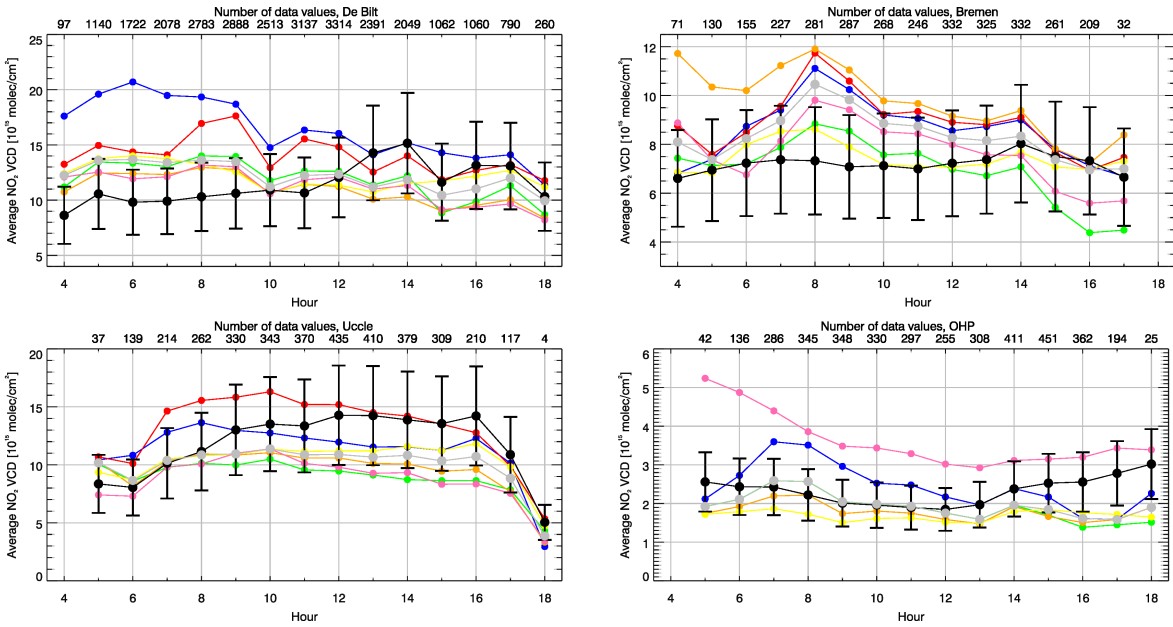

**Figure 9.** As in Figure 8 but for diurnal cycles (averages over hourly bins) of tropospheric $NO_2$ VCDs [$10^{15}$ molec cm$^{-2}$].

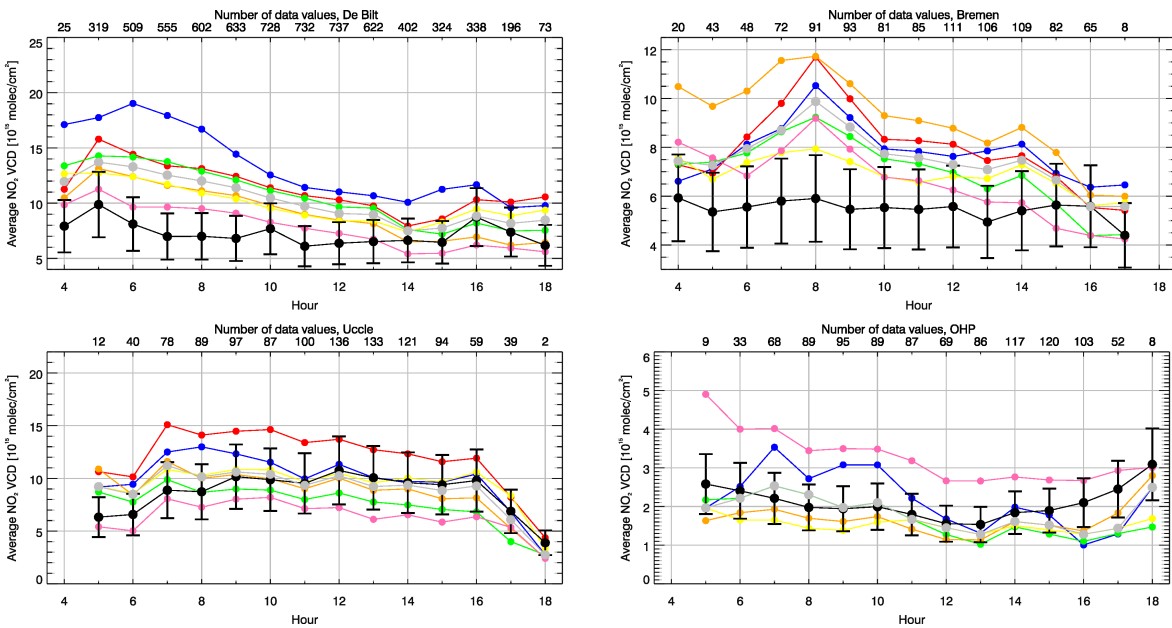

**Figure 10.** As in Figure 8 but for diurnal cycles (averages over hourly bins) of tropospheric $NO_2$ VCDs [$10^{15}$ molec cm$^{-2}$] during weekends only.

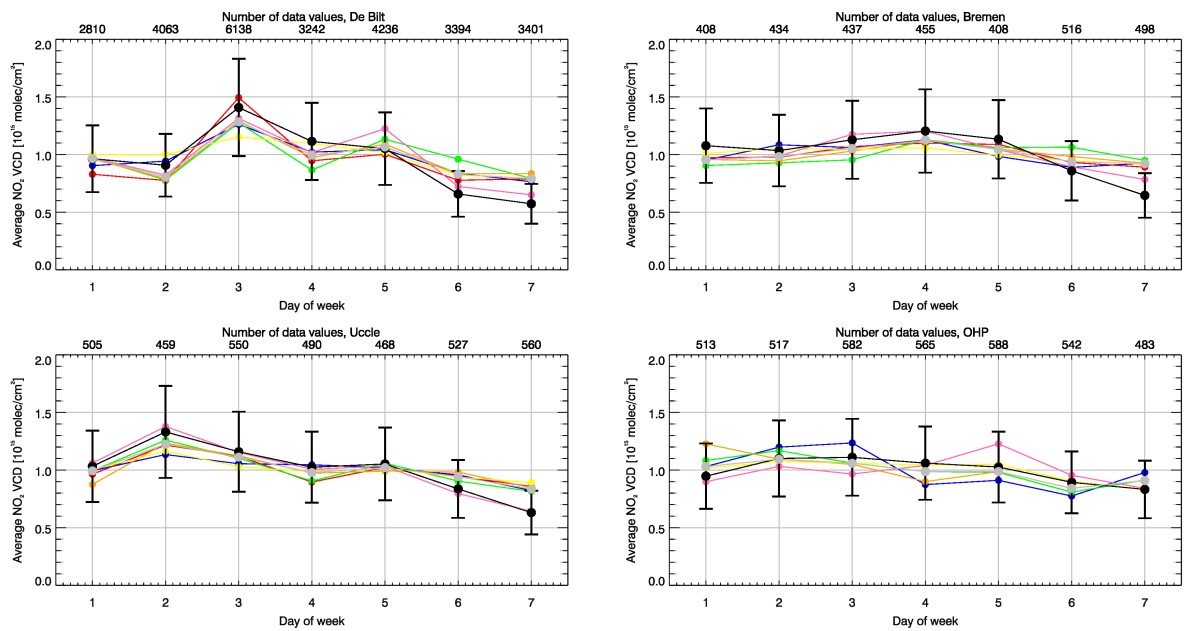

**Figure 11.** As in Figure 8 but for weekly cycles (averages over daily bins devided by mean over whole week, unitless values) of tropospheric NO$_2$ VCDs.

**Table 1.** Overview of regional air quality model simulations.

| Model | Institution | Grid spacing (zonal x meridional) | Number of vertical levels model top | Chemistry scheme |
|---|---|---|---|---|
| CHIMERE | LISA-CNRS/UPEC/UPD INERIS | 0.25° x 0.25° ($\sim$ 18 x 28 km$^2$) | 8 500 hPa | MELCHIOR II (Schmidt et al., 2001) |
| EMEP-MACCEVA | MetNo | 0.25° x 0.125° ($\sim$ 18 x 14 km$^2$) | 20 100 hPa | EMEP-EmChem09soa (Simpson et al., 2012; Bergström et al., 2012) |
| EMEP | MetNo | 50 x 50 km$^2$ | 20 100 hPa | EMEP-EmChem09soa (Simpson et al., 2012 Bergström et al., 2012) |
| LOTOS-EUROS | TNO | 0.125° x 0.0625° ($\sim$ 9 x 7 km$^2$) | 3 $\sim$ 3.5 km | TNO CBM-IV (Schaap et al., 2008; Whitten et al., 1980) |
| MOCAGE | CNRS-Météo-France | 0.2° x 0.2° ($\sim$ 15 x 22 km$^2$) | 47 5 hPa | troposphere: RACM (Stockwell et al., 1997) stratosphere REPROBUS (Lefèvre et al., 1994) |
| SILAM | FMI | years 2010/2011: 0.2° x 0.2° ($\sim$ 15 x 22 km$^2$) year 2012: 0.15° x 0.15° ($\sim$ 11 x 17 km$^2$) | 9 (2010), 8 (2011-2012) 6.725 km (2010), 6.7 km (2011-2012) | DMAT (Sofiev, 2000) |

**Table 2.** Overview of MAX-DOAS station data.

| Station location | lat, lon height [masl] | Institution | Time period | Type | Retrieved quantitiy | number of layers layer top [km] | additional data |
|---|---|---|---|---|---|---|---|
| De Bilt Netherlands | 52.1° N, 5.18° E $\sim$ 23 m | KNMI | 03/2011–12/2012 | urban | column | 12 4.0 km | wind data (in-situ) |
| Bremen Germany | 53.11° N, 8.86° E 21 m | IUP-UB | 01/2011–12/2011 | urban | column profile | 81 4.025 km | wind data (in-situ data from airport weather station $\sim$ 9 km southwards at 53.05° N, 8.79° E) |
| Uccle Belgium | 50.8° N, 4.32° E 120 m | BIRA-IASB | 01/2012–12/2012 | urban | column profile | 13 3.5 km | wind data (in-situ) clouds from MAX-DOAS |
| OHP France | 43.92° N, 5.7° E 650 m | BIRA-IASB | 01/2010-12/2010 | rural | column profile | 13 3.5 km | wind data (in-situ) |

**Table 3.** Statistics on how AVK-weighted tropospheric $NO_2$ VCDs [$10^{15}$ molec cm$^{-2}$] from regional models compare to MAX-DOAS retrievals at the four MAX-DOAS stations. Each column entry shows from left to right: root mean squared error [$10^{15}$ molec cm$^{-2}$], bias [$10^{15}$ molec cm$^{-2}$] and Pearson correlation coefficient. MOCAGE data is not available for the measurement time period at OHP.

| | De Bilt | | | Bremen | | | Uccle | | | OHP | | |
|---|---|---|---|---|---|---|---|---|---|---|---|---|
| | rms | bias | r | rms | bias | r | rms | bias | r | rms | bias | r |
| ENSEMBLE | 7.955 | 0.833 | 0.465 | 4.142 | 0.898 | 0.583 | 6.862 | -2.220 | 0.719 | 1.369 | -0.341 | 0.683 |
| LOTOS-EUROS | 10.516 | 5.254 | 0.352 | 5.180 | 1.598 | 0.461 | 7.815 | -0.897 | 0.598 | 3.004 | 0.217 | 0.666 |
| CHIMERE | 8.221 | 0.573 | 0.402 | 3.960 | 0.198 | 0.533 | 7.059 | -1.950 | 0.686 | 1.269 | -0.563 | 0.627 |
| EMEP | 8.680 | 0.919 | 0.393 | 4.558 | -0.167 | 0.521 | 8.134 | -3.609 | 0.624 | 1.824 | -0.338 | 0.600 |
| EMEP-MACCEVA | 8.340 | -0.182 | 0.397 | 5.308 | 2.427 | 0.554 | 7.591 | -2.777 | 0.659 | 2.012 | -0.473 | 0.532 |
| SILAM | 8.516 | 0.115 | 0.408 | 4.506 | 0.570 | 0.550 | 7.985 | -3.385 | 0.633 | 2.577 | 1.195 | 0.482 |
| MOCAGE | 9.731 | 3.082 | 0.427 | 5.651 | 1.801 | 0.520 | 7.413 | 1.476 | 0.692 | | | |

**Table 4.** As in Table 3 but for $NO_2$ surface partial columns [$10^{15}$ molec cm$^{-2}$]. Surface partial columns from MAX-DOAS are not available for De Bilt for the investigated time period.

| | Bremen | | | Uccle | | | OHP | | |
|---|---|---|---|---|---|---|---|---|---|
| | rms | bias | r | rms | bias | r | rms | bias | r |
| ENSEMBLE | 0.715 | 0.123 | 0.374 | 2.905 | -1.124 | 0.586 | 0.351 | 0.058 | 0.439 |
| LOTOS-EUROS | 0.783 | 0.181 | 0.336 | 3.309 | -1.659 | 0.509 | 0.517 | 0.048 | 0.393 |
| CHIMERE | 0.927 | 0.364 | 0.252 | 2.902 | -0.554 | 0.531 | 0.337 | 0.081 | 0.400 |
| EMEP | 0.723 | -0.133 | 0.318 | 3.330 | -1.819 | 0.533 | 0.443 | 0.068 | 0.417 |
| EMEP-MACCEVA | 0.869 | 0.414 | 0.320 | 3.229 | -1.658 | 0.548 | 0.428 | 0.014 | 0.344 |
| SILAM | 0.681 | -0.054 | 0.397 | 2.910 | -0.498 | 0.572 | 0.659 | 0.388 | 0.318 |
| MOCAGE | 0.750 | 0.101 | 0.372 | 2.886 | 0.272 | 0.596 | | | |

**Table 5.** As in Table 3 but for (upper rows) seasonal cycles, (middle rows) diurnal cycles and (lower rows) weekly cycles of AVK-weighted tropospheric NO$_2$ VCDs [$10^{15}$ molec cm$^{-2}$]. In addition, values for diurnal cycles are given based on data during weekdays only and during weekends only for the ensemble.

| | | De Bilt | | | Bremen | | | Uccle | | | OHP | | |
|---|---|---|---|---|---|---|---|---|---|---|---|---|---|
| | | rms | bias | r | rms | bias | r | rms | bias | r | rms | bias | r |
| seasonal | ENSEMBLE | 3.750 | 2.494 | 0.511 | 2.218 | 1.347 | 0.710 | 2.857 | -2.074 | 0.863 | 1.055 | 0.018 | 0.755 |
| | LOTOS-EUROS | 6.982 | 5.920 | 0.413 | 2.808 | 1.807 | 0.708 | 2.648 | -0.892 | 0.777 | 1.975 | 0.752 | 0.859 |
| | CHIMERE | 4.001 | 2.592 | 0.343 | 1.291 | 0.519 | 0.627 | 3.337 | -1.853 | 0.810 | 0.563 | -0.507 | 0.878 |
| | EMEP | 3.237 | 1.837 | 0.575 | 1.960 | 0.337 | 0.743 | 4.000 | -3.071 | 0.820 | 1.187 | -0.020 | 0.733 |
| | EMEP-MACCEVA | 3.647 | 1.587 | 0.304 | 3.344 | 2.595 | 0.702 | 3.455 | -2.581 | 0.832 | 0.903 | -0.208 | 0.795 |
| | SILAM | 2.904 | 1.631 | 0.617 | 1.933 | 0.905 | 0.688 | 3.323 | -2.759 | 0.853 | 1.506 | 1.270 | 0.798 |
| | MOCAGE | 7.443 | 5.315 | 0.214 | 3.823 | 2.414 | 0.643 | 2.621 | 1.594 | 0.853 | | | |
| diurnal | ENSEMBLE | 2.571 | 0.749 | -0.389 | 1.499 | 1.134 | 0.148 | 2.401 | -1.735 | 0.815 | 0.599 | -0.371 | -0.069 |
| | ENSEMBLE (weekdays) | 2.762 | 0.103 | -0.237 | 1.343 | 0.752 | 0.125 | 3.181 | -2.527 | 0.827 | 0.639 | -0.390 | -0.251 |
| | ENSEMBLE (weekends) | 3.438 | 3.006 | 0.559 | 2.664 | 2.463 | 0.405 | 1.294 | 0.341 | 0.806 | 0.461 | -0.253 | 0.567 |
| | LOTOS-EUROS | 6.283 | 4.959 | -0.525 | 1.777 | 1.362 | 0.413 | 2.055 | -0.448 | 0.715 | 0.756 | 0.106 | -0.255 |
| | CHIMERE | 2.560 | 0.813 | -0.456 | 0.614 | 0.250 | 0.335 | 1.927 | -1.379 | 0.919 | 0.703 | -0.641 | 0.585 |
| | EMEP | 2.690 | 0.482 | -0.219 | 1.339 | -0.251 | 0.052 | 3.568 | -2.739 | 0.567 | 0.711 | -0.435 | -0.298 |
| | EMEP-MACCEVA | 2.681 | -0.450 | -0.424 | 2.971 | 2.574 | -0.235 | 2.911 | -2.194 | 0.724 | 0.662 | -0.540 | 0.128 |
| | SILAM | 2.620 | -0.199 | -0.327 | 1.429 | 0.497 | -0.085 | 3.410 | -2.895 | 0.763 | 1.468 | 1.301 | 0.270 |
| | MOCAGE | 3.782 | 2.639 | -0.262 | 2.070 | 1.670 | 0.149 | 2.277 | 1.433 | 0.812 | | | |
| weekly | ENSEMBLE | 0.129 | 0.009 | 0.917 | 0.131 | -0.009 | 0.820 | 0.101 | 0.006 | 0.967 | 0.060 | -0.009 | 0.802 |
| | LOTOS-EUROS | 0.122 | 0.014 | 0.970 | 0.136 | -0.008 | 0.699 | 0.122 | -0.005 | 0.973 | 0.129 | 0.005 | 0.572 |
| | CHIMERE | 0.147 | 0.029 | 0.973 | 0.138 | -0.009 | 0.878 | 0.134 | -0.006 | 0.954 | 0.036 | 0.001 | 0.935 |
| | EMEP | 0.186 | 0.014 | 0.708 | 0.176 | -0.013 | 0.278 | 0.095 | -0.004 | 0.928 | 0.082 | 0.004 | 0.696 |
| | EMEP-MACCEVA | 0.146 | 0.014 | 0.865 | 0.141 | -0.010 | 0.727 | 0.126 | -0.006 | 0.828 | 0.130 | 0.005 | 0.335 |
| | SILAM | 0.097 | 0.007 | 0.930 | 0.083 | -0.006 | 0.898 | 0.028 | 0.000 | 0.993 | 0.102 | -0.001 | 0.557 |
| | MOCAGE | 0.139 | -0.009 | 0.850 | 0.117 | -0.008 | 0.899 | 0.120 | -0.006 | 0.872 | | | |

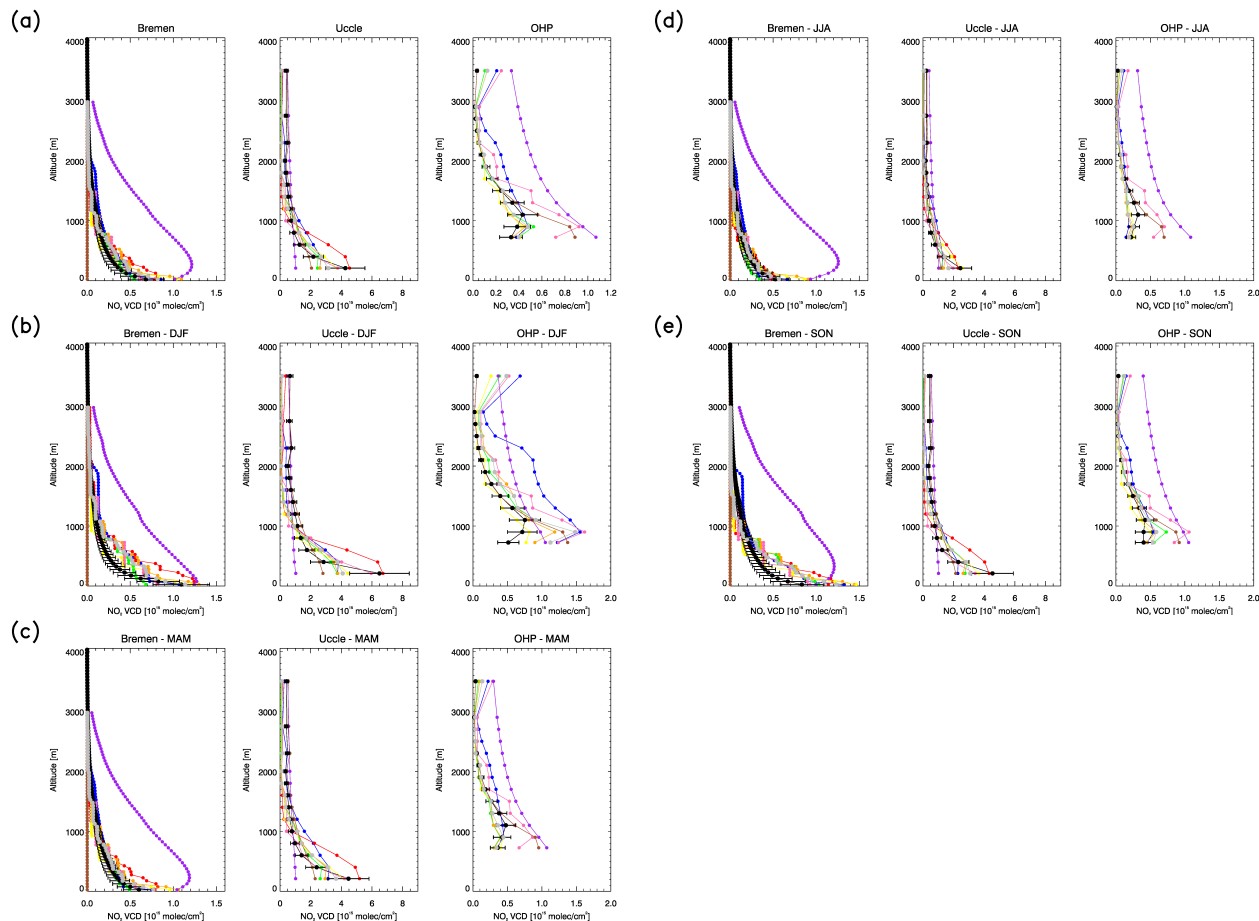

**Figure A1.** Average vertical profiles of $NO_2$ partial columns [$10^{15}$ molec cm$^{-2}$] from (black) MAX-DOAS, (brown) a priori used for MAX-DOAS retrievals, (gray) model ensemble median, (blue) LOTOS-EUROS, (yellow) CHIMERE, (green) EMEP, (orange) EMEP-MACCEVA, (pink) SILAM and (red) MOCAGE. Black error bars refer to the uncertainty associated with the MAX-DOAS retrievals (assumed to be 30 % for all stations). Panel (a) shows profiles for data averaged over whole time series, panel (b) shows profiles for DJF (December, January February), (c) MAM (March, April, May), (d) JJA (June, July, August) and (e) SON (September, October, November) months only. Figures in panels (a) to (e) refer to (left) Bremen, (middle) Uccle and (right) OHP. MAX-DOAS vertical profiles are not available for De Bilt for the investigated time period.

## Appendix A:  Appendix

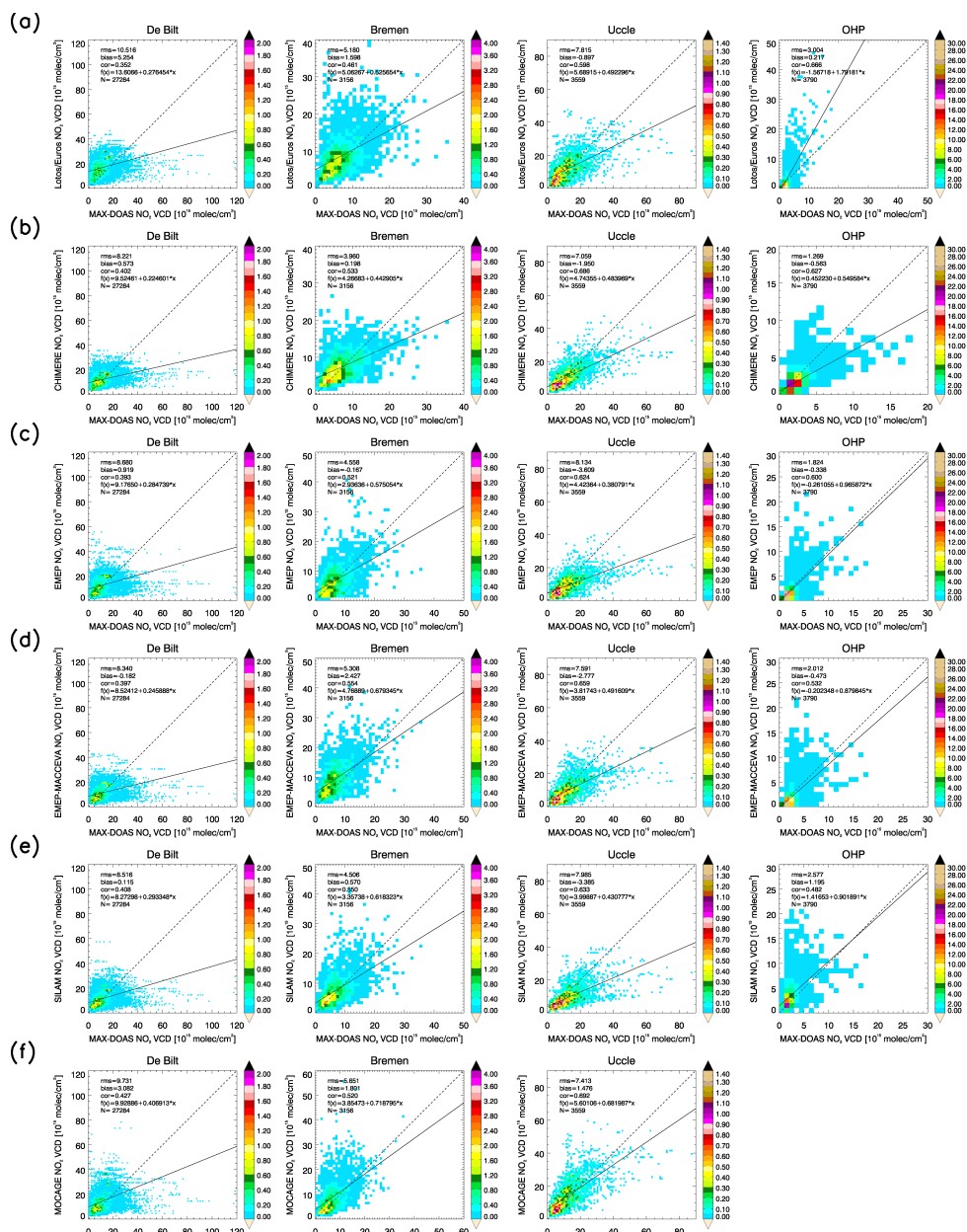

**Figure A2.** Scatter density plots of AVK-weighted tropospheric $NO_2$ VCDs [$10^{15}$ molec cm$^{-2}$] from MAX-DOAS against model data for (from left to right) De Bilt, Bremen, Uccle and OHP. The data is shown for different bins with a size of $10^{15}$ molec cm$^{-2}$ and is colored according to the number of data points per bin [%].The different panels show different model runs: (a) LOTOS-EUROS, (b) CHIMERE, (c) EMEP, (d) EMEP-MACCEVA, (e) SILAM and (f) MOCAGE. MOCAGE data is not available for the measurement time period at OHP. The dashed line is the reference line (f(x)=x). The solid line is the regression line (see top left of each plot for f(x) of this line). The root mean squared error (rms) [$10^{15}$ molec cm$^{-2}$], bias [$10^{15}$ molec cm$^{-2}$], Pearson correlation coefficient (cor, not squared) as well as the number of data points N are given at the top left of each plot.

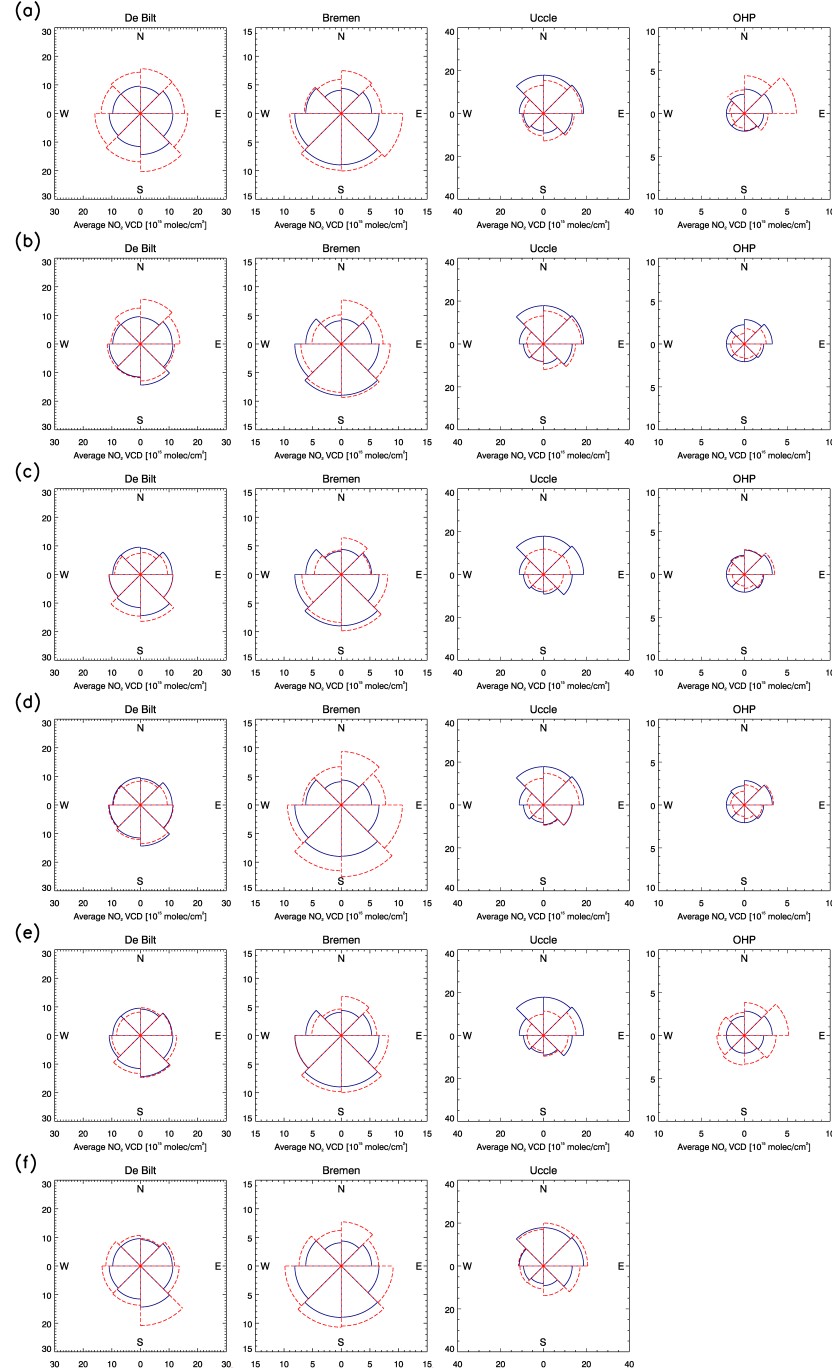

**Figure A3.** As in Figure A2 but for average AVK-weighted tropospheric $NO_2$ VCDs [$10^{15}$ molec cm$^{-2}$] in 45° wide wind direction bins from (blue solid lines) MAX-DOAS and (red dashed lines) model data calculated. MOCAGE data is not available for the measurement time period at OHP.

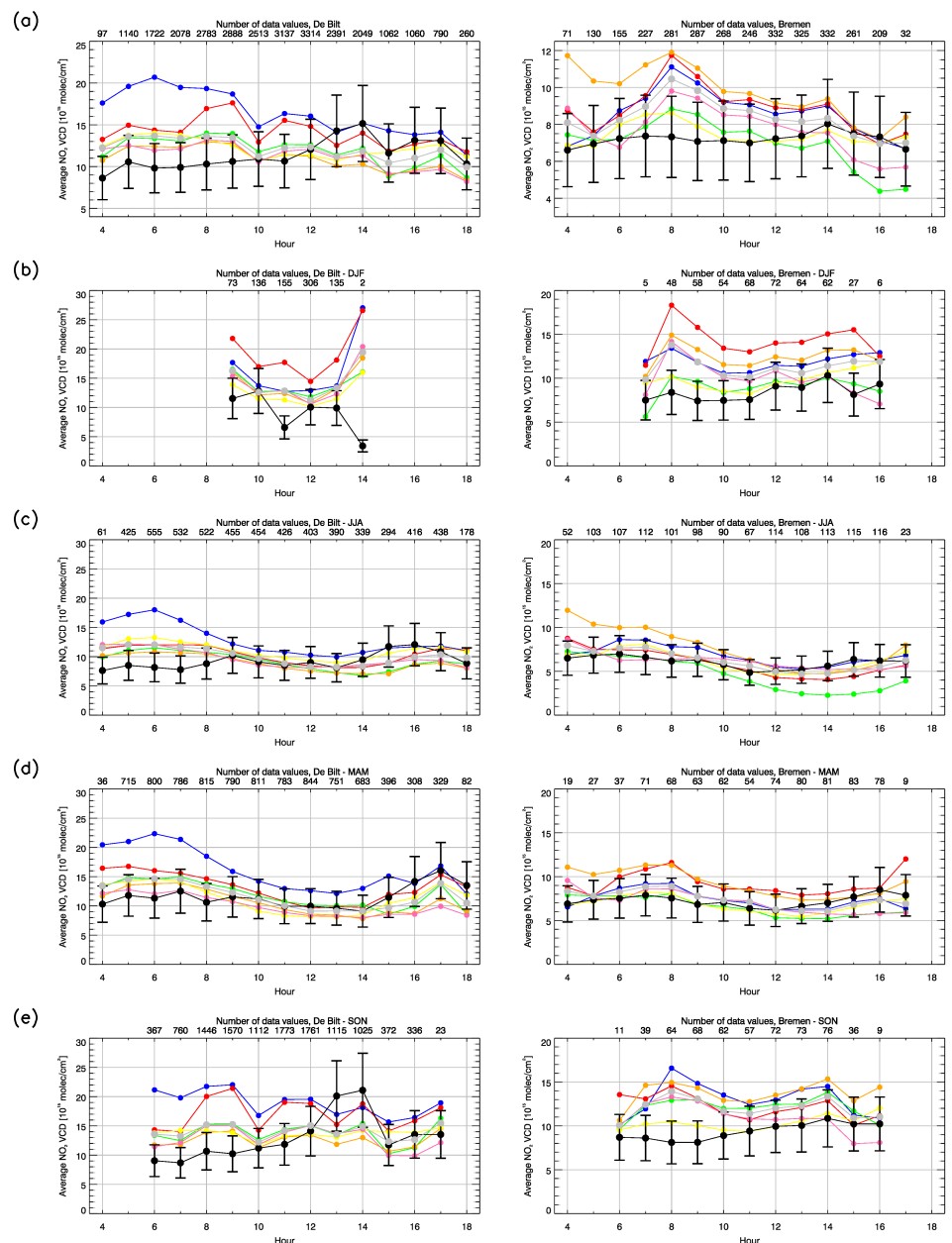

**Figure A4.** Diurnal cycles (averages over hourly bins) of AVK-weighted tropospheric NO$_2$ VCDs [$10^{15}$ molec cm$^{-2}$] from (black) MAX-DOAS, (gray) model ensemble median, (blue) LOTOS-EUROS, (yellow) CHIMERE, (green) EMEP, (orange) EMEP-MACCEVA, (pink) SILAM and (red) MOCAGE for (left) De Bilt and (right) Bremen. Model based diurnal cycles were calculated from tropospheric NO$_2$ VCDs. Black error bars refer to the uncertainty associated with the MAX-DOAS retrievals (assumed to be 30 % for all stations). Panel (a) shows cycles for the whole time series, panel (b) shows cycles for DJF, (c) MAM, (d) JJA and (e) SON months only. The number of data values used for calculating average values is shown at the top x-axis of each plot.

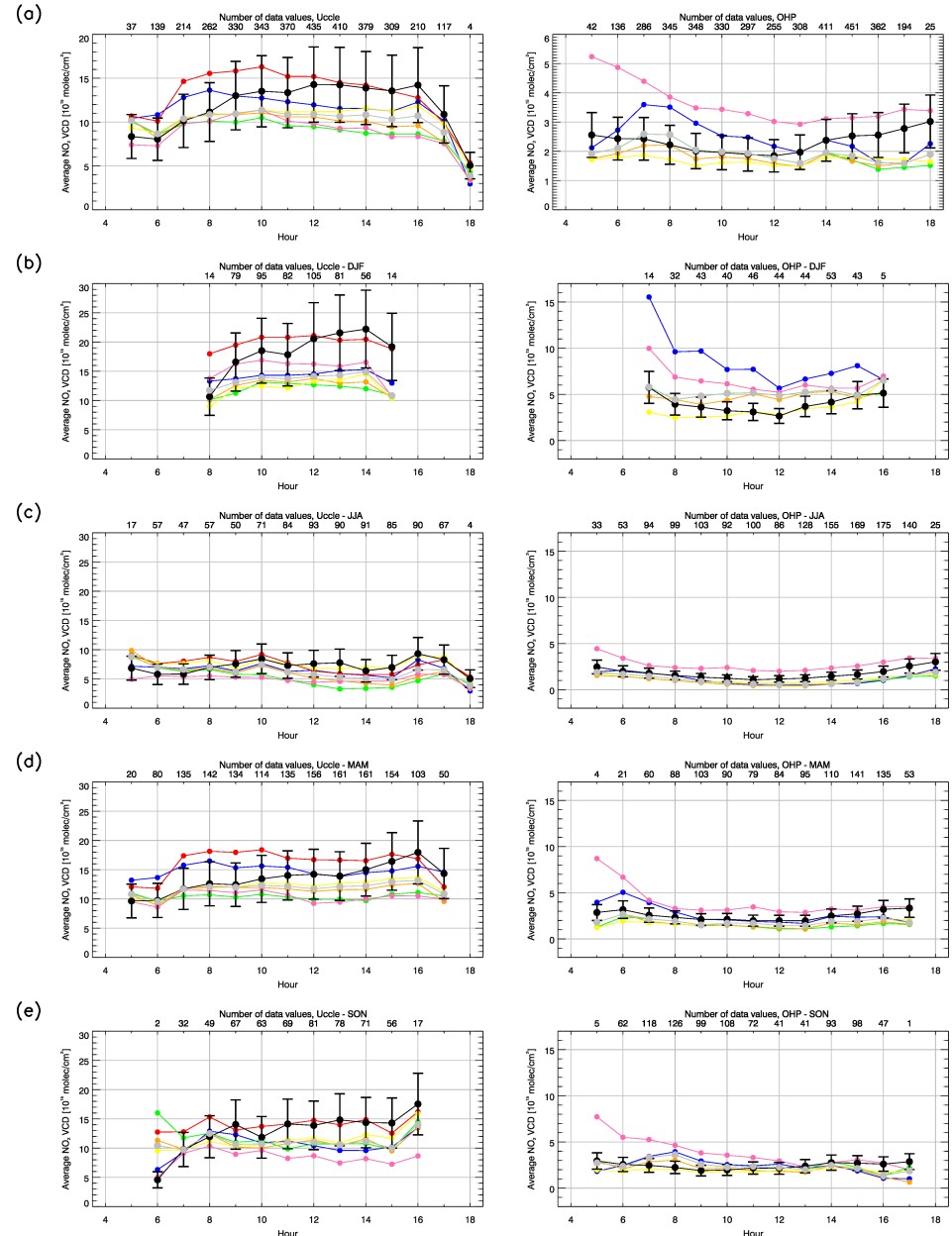

**Figure A5.** As in Figure A4 but for (left) Uccle and (right) OHP.

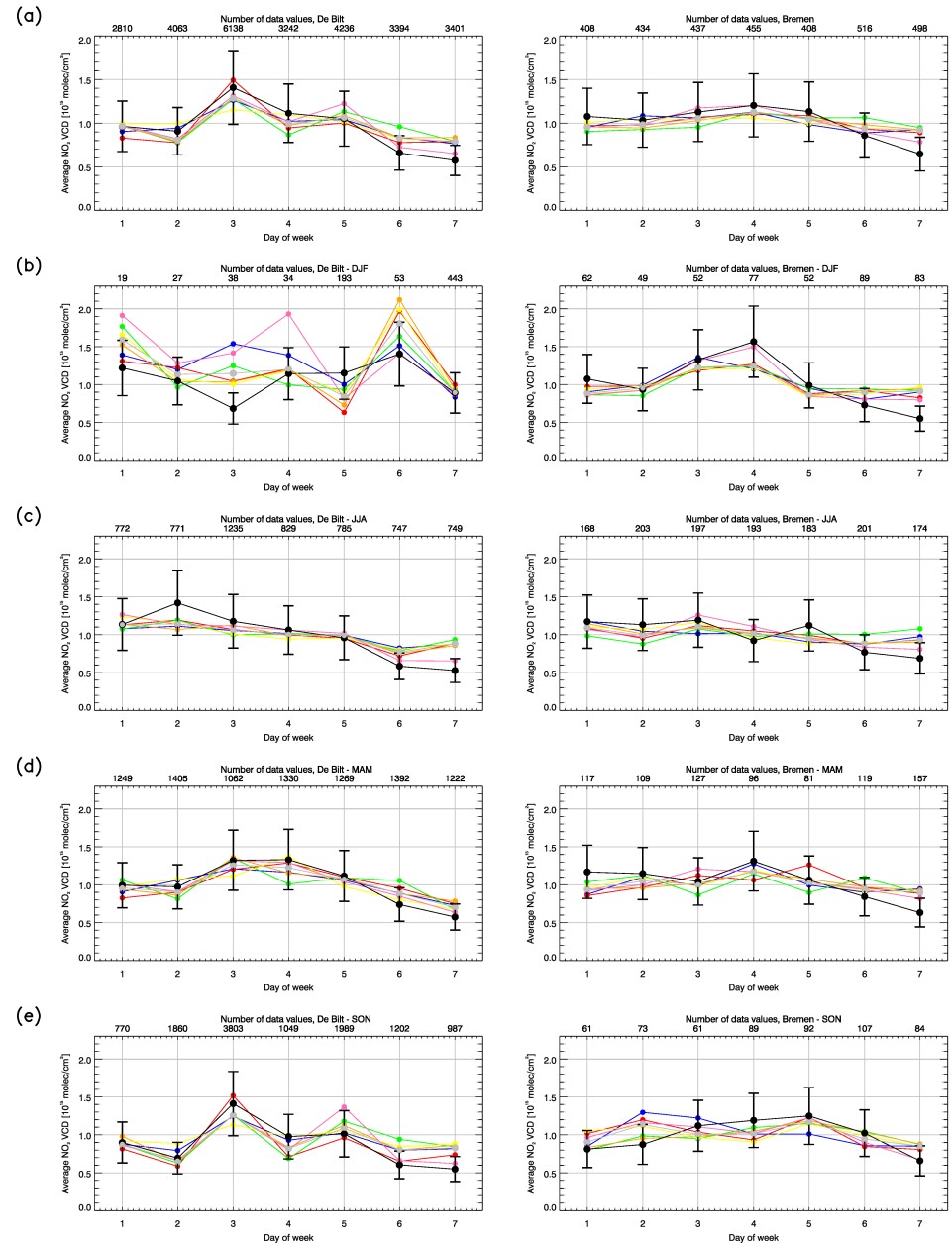

**Figure A6.** As in Figure A4 but for weekly cycles (averages over daily bins devided by mean over whole week, unitless values) of tropospheric $NO_2$ VCDs.

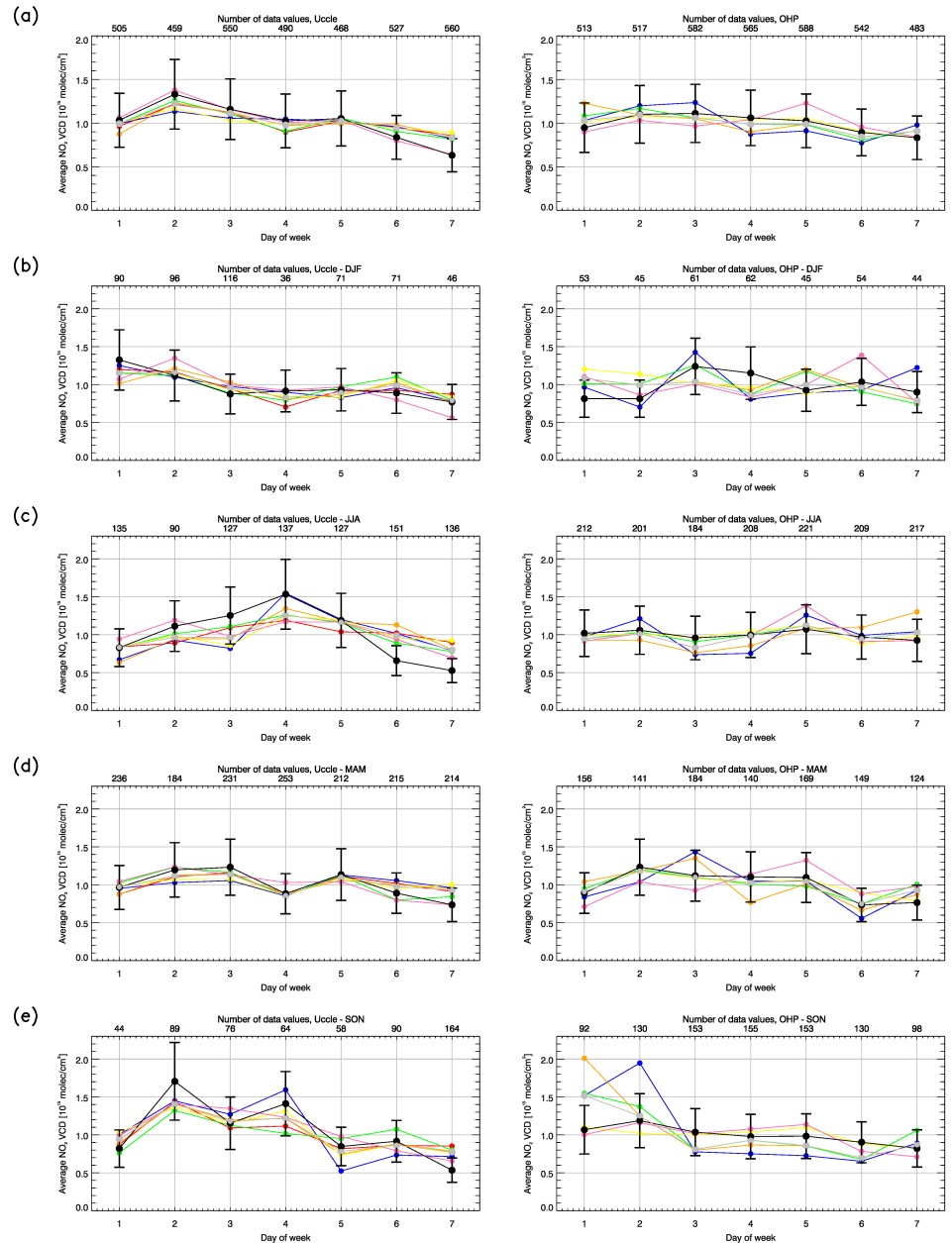

**Figure A7.** As in Figure A5 but for weekly cycles (averages over daily bins devided by mean over whole week, unitless values) of tropospheric $NO_2$ VCDs.