# Peer review of "Comparison of tropospheric NO2 columns from MAX-DOAS retrievals and regional air quality model simulations"

_Atmospheric Chemistry and Physics, 2016_

## Referee Comment (RC1) · Anonymous Referee #3 · 16 Mar 2017

The topic is very relevant, however I have to criticise the approach used since in the current status important open questions remain.

Let me start by asking the author, once more, to improve the language adopted in the manuscript. There are some fixed points on which the community has agreed upon since many years that simply cannot be ignored. For example the term 'validation' should be dropped for the time being in favour of 'evaluation'. This has been clearly stated in a number of important publications that cannot be neglected. Secondly, one cannot talk about 'validatio'n and than start two sections with "Intercomparison method" and "Intercomparison results". A comparison is between two or more things normally. The suffix 'inter' normally refers to a comparison of elements of the same nature, e.g.

model vs model, obs vs obs. If that would not be the case, one would simply talk about "comparison", would he/she not? I think that the natures of observation and model results are already sufficiently different, to complicate further the scene and inferring, with the used of 'intercomparison', that they are not. How about "methodology for the evaluation of the ensemble" and "Results", simple straight forward, clear?

I have serious problems with reading the figures. They are excruciatingly small and the number and nature of the differences between models and models/obs is so crucial to the evaluation of the manuscript quality that I cannot precede in a conclusive way.

The most important objection resides in the ensemble treatment and the fact that the differences among the models are confined in the appendix of the paper. The differences among the models qualify the ensemble and define also the quality of your final results. Once more the figures are too small for me to say something definitive here, but from what I can judge I see small differences among models. This puts in question the necessity for an ensemble treatment especially when based on the median which by definition cuts the outliers contribution to the ensemble result and in this particular case may well make redundant the use of several models that are replicating their results. May be they are different, but this is not visible to me from the figures provided.

I do know the value of ensembles of opportunity but the opportunity should be exploited at maximum making sure that there is an added value within the use of multiple models, that the number of models is adequate, not too many not too few and that the contribution from the model results finally used is original and un biased. This has been demonstrated in a number of works that deserve the attention of the authors.

I think the paper will benefit if the individual relationships among the ensemble members is brought to a higher degree of visibility (not only with larger figures but also conceptually) and analysis. This will increase the scientific significance of this paper which otherwise would look too much like a performance report. The later is useful indeed for the institution/s that use these results but is not at all instructive for the

scientific community.

What I find contradicting a bit in this paper is also the fact that data are used to validate an ensemble, use nature is obscure, and the main message that this brought forward is indirectly that this exercise demonstrates that Max-Doas data are suitable to validate models. So what is validating what and how?

In the present status the manuscript can not, in my view be published in ACP. GMD would be more suitable, but provided that more insight is given into the ensemble workings.

I am prepared to read a new version of this work should the Editor consider it necessary.

---

## Referee Comment (RC2) · Anonymous Referee #1 · 24 Mar 2017

The paper presents a comparison of time series of tropospheric NO2 VCDs derived from 4 European MAX-DOAS stations to an ensemble of 5 regional models. The horizontal and vertical resolution of MAX-DOAS observations fits in general well to those of the regional models. Thus such a comparison is well suited to evaluate the performance of the model simulations (and also the quality of the MAX-DOAS retrievals). In this respect, the results of this paper are of high importance, and are well suited for publication in ACP. However, I have three major concerns with respect to the evaluation and presentation of the results in the present version of the manuscript, which should be addressed before final publication:

a) One of the main advantages of MAX-DOAS observations is that profile information

for the lowest layers of the atmosphere (below about 2km) can be obtained. Profile information is crucial to assess the performance of the model simulations (and to understand deviations from observations). It is a pity (and completely unclear to me), why the authors do not make explicit use of the profile information derived from MAX-DOAS. One – rather simple – way to make use of the profile information (and to compare MAX-DOAS results and model simulations) would be to determine a characteristic layer height (e.g. the layer, below 70% of the total tropospheric column resides) from both the MAX-DOAS observations and the model results.

b) The authors compare the MAX-DOAS results to model ensembles. Although in the appendix, also the comparison results to the individual models are shown, no attempt is made to systematically asses the performance of the individual models with respect to the MAX-DOAS results. The authors should at least provide a table with some key indicators (e.g. correlation coefficient, slope, bias, etc.) for the individual model comparisons. These indicators should be provided for a) the complete time series, b) for the seasonal variation, c) the diurnal variation, and d) the weekly cycle.

c) The discussion of the deviations between the model simulations and the MAX-DOAS results is weak, and only rather general explanations for the disagreements are given. The paper would benefit a lot if the possible reasons for disagreement would be investigated in more depth. In particular, from the two points mentioned above, useful information could be obtained, which processes (e.g. transport, emission inventories, chemistry) might be most important reason for discrepancies for individual situations and/or models.

Minor points:

Page 1, line 1: Replace NO2 by NOx

Page 1, line 8: 'measurements are available during daylight'. To me it seems that this is not an advantage but rather a disadvantage (measurements are not available during night)

Introduction: It should be made more clear, that the quantity of interest is NOx, but only NO2 can be measured

Page 2, line 30: The statement 'using zenith measurements as intensity of incident radiation' is unclear to me. Do you mean incident solar irradiation? Then I would disagree. Please clarify.

Section 2.1: What is the spatial resolution of the models? How does it compare to the horizontal sensitivity ranges of the MAX-DOAS results?

Section 2.2: The retrievals are described in an inconsistent and partly incomplete way. For example, for KNMI the retrieval procedure is completely unclear. Was a profile inversion performed or not? This section should be harmonised and completed. The effect of the different inversion procedures on the NO2 results should be briefly discussed.

Section 2.2: It is stated that for Uccle, cloud information was retrieved. Was this information also used for the selection of the measurements? What about the retrieval of cloud information for the other stations?

Section 2.3: How does the wind data compare to the wind fields used in the models? What about wind data for KNMI?

Page 8, line 22: 'Only those model values closest to the measurement time are used'. Why is no interpolation in time of neighbouring model output values performed?

Page 9, line 10: What is the vertical extension of the lowest measurement layer?

Page 9, line 12: 'comparisons of profiles'? No comparison of profiles is shown in Figs. 1 and 2.

Page 10, line 5: 'As the sensitivity of MAX-DOAS retrievals is largest in the boundary layer' Is this also true for the 'de Bilt measurements'?

Page 10, lines 23,24: 'On average, observed NO2 partial columns are higher in the

lowest observation layers during cloudy conditions compared to clear-sky conditions' I guess that no clouds are considered in the MAX-DOAS forward model. How reliable are then the MAX-DOAS NO2 results under cloudy conditions?

Page 11, line 3: What is exactly meant with 'correlation'? r or r squared?

Page 11, line 12: How consistent are the wind data from the weather stations with the wind fields used in the models? Can you show a similar plot as Fig. 6 based on the wind fields from the models?

Page 13, line 15: 'However, many validation points arise from the MAX-DOAS based comparisons which could improve model performance substantially.' This sentence is not clear to me. Please clarify.

---

## Referee Comment (RC3) · Anonymous Referee #2 · 28 Mar 2017

In "Comparison of tropospheric NO2 columns from MAX-DOAS retrievals and regional air quality model simulations," the authors provide a nice overview of 1) long-term MAX-DOAS records of NO2 in northwest and southwest Europe, 2) a description of regional air quality models used in the CAMS ensemble, and 3) a description of past comparisons of regional CTMs and MAX-DOAS with in situ and satellite data. The comparison of the model ensemble and the four MAX-DOAS NO2 datasets showed general agreement in a broad sense. The authors highlight when and where there are discrepancies between ensemble median model results and MAX-DOAS observations (e.g., seasonal cycle, diurnal cycle), but do not offer ideas on potential approaches for disentangling the causes of these discrepancies.

[Figure]

I felt that the paper lacked a final synthesis, written in more general language, of how future simulations and MAX-DOAS deployments like these can isolate effects from individual processes. I hope that the authors consider adding a broader synthesis of their results to the end of section 4, offering possible paths forward for future analyses: what common and distinct attributes of these four sites share? How might these differences and similarities be exploited to investigate chemistry? Emissions? Meteorology? Where might the authors propose future MAX-DOAS instruments be located? Should one expect an ensemble median to capture hourly $NO_2$ variations? Monthly averages? What is the native scale of $NO_2$ spatial variations at the MAX-DOAS sites inferred from the time scale of $NO_2$ variation and wind speed?

Comments: P2, L10-11: $NO_2$ lifetime is much longer in the upper troposphere, primarily because its chemical family, $NO_x$, is mostly present as NO at high altitudes, which has far fewer permanent sinks.

P3, L29: "focusses" – typo

P4: There is no discussion of model resolution. The $NO_2$ lifetime is a function of model resolution. Also, median values may be biased towards coarser models as those with finer resolution may produce highs when a plume passes and lows when not.

P5, L7: These sites, with exception of OHP, seem to be in very similar physical settings, with likely similar meteorology (e.g., vertical mixing characteristics). If so, this fact should be mentioned. Please also consider including a map of the region with sites indicated on a backdrop of satellite-based tropospheric $NO_2$ column measurements. Minor comment: I did not see Lat/Lon values reported for Uccle.

Page 6, Line 29: Has there been any side-by-side operation and comparison of these two instruments? If so, please provide the reference.

P9, L5: "As the typical error on MAX-DOAS retrieved VCDs is around 20%" – please describe this statement in more detail: at what time scale? Random or systematic

uncertainty? Based on measurement inter-comparisons or fitting statistics? Page 9, L17-21: See comment on "page 4" above. NOx lifetime depends on model resolution, and NO2 maxima will be diluted in coarser models. Model resolution needs to be better reported.

P10, L5-17: I have a hard time following the language and reasoning behind this conclusion. Please consider clarifying. Is this because the a priori profiles are generated from similar models as those included in the comparison? Are any systematic effects buried below random sources of uncertainty?

Page 10, L21-31: This analysis and discussion is tangential to the broader scope of the paper and should be removed, as earlier noted by the authors "The impact of clouds on MAX-DOAS retrievals is described in detail by Vlemmix et al. (2015)" I do consider the comparison of model and MAX-DOAS NO2 columns under different cloud conditions to be an interesting topic for its own manuscript.

P10-11, L34-11: How much of the correlation is determined by seasonal and weekly cycle? Consider isolating correlation at one time of day, one season and one set of weekdays (e.g., M-F)

P12, L35: Consider a reference to Beirle et al. (2003). I think that this paragraph could be expanded. Day-of-week effects, over the long-term, are independent of meteorology and driven entirely by variations of emissions and chemistry. Future day-of-week comparisons would be one means of providing systematic approaches to quantify the many processes affecting NO2 (emissions, meteorology uncertainty, chemistry, observational uncertainty)
* * *

---

## Author Comment (AC1) · 6 Sep 2018

**Response to anonymous referee #3:**

**We thank referee #3 for constructive and helpful review comments, to which we hope to have responded appropriately. A list of comments including our response is given below.**

*The topic is very relevant, however I have to criticise the approach used since in the current status important open questions remain.*

*Let me start by asking the author, once more, to improve the language adopted in the manuscript. There are some fixed points on which the community has agreed upon since many years that simply cannot be ignored. For example the term 'validation' should be dropped for the time being in favour of 'evaluation'. This has been clearly stated in a number of important publications that cannot be neglected. Secondly, one cannot talk about 'validatio'n and than start two sections with "Intercomparison method" and "Intercomparison results". A comparison is between two or more things normally. The suffix 'inter' normally refers to a comparison of elements of the same nature, e.g. model vs model, obs vs obs. If that would not be the case, one would simply talk about "comparison", would he/she not? I think that the natures of observation and model results are already sufficiently different, to complicate further the scene and inferring, with the used of 'intercomparison', that they are not. How about "methodology for the evaluation of the ensemble" and "Results", simple straight forward, clear?*

We apologize in case the terms „validation" and „intercomparison" were still used in an inappropriate manner, this was not intended. The corresponding text has been changed as suggested.

*I have serious problems with reading the figures. They are excruciatingly small and the number and nature of the differences between models and models/obs is so crucial to the evaluation of the manuscript quality that I cannot precede in a conclusive way.*

*The most important objection resides in the ensemble treatment and the fact that the differences among the models are confined in the appendix of the paper. The differences among the models qualify the ensemble and define also the quality of your final results. Once more the figures are too small for me to say something definitive here, but from what I can judge I see small differences among models. This puts in question the necessity for an ensemble treatment especially when based on the median which by definition cuts the outliers contribution to the ensemble result and in this particular case may well make redundant the use of several models that are replicating their results. May be they are different, but this is not visible to me from the figures provided.*
*I do know the value of ensembles of opportunity but the opportunity should be exploited at maximum making sure that there is an added value within the use of multiple models, that the number of models is adequate, not too many not too few and that the contribution from the model results finally used is original and unbiased. This has been demonstrated in a number of works that deserve the attention of the authors.*

*I think the paper will benefit if the individual relationships among the ensemble members is brought to a higher degree of visibility (not only with larger figures but also conceptually) and analysis. This will increase the scientific significance of this paper which otherwise would look too much like a performance report. The later is useful indeed for the institution/s that use these results but is not at all instructive for the scientific community.*

Many changes have been applied to Figures and Tables in order to increase visibility of individual model runs and to enlarge Figures shown in the main part of the manuscript::

-Figures showing non AVK-weighted tropospheric $NO_2$ VCDs (termed tropospheric $NO_2$ VCDs from method 1 in previous version) were deleted as these do not differ substantially from AVK-weighted (referred to as method 2 in previous version) values (see p 11 l 19 - p 12 l 2, revised version).

-Scatter density plots and wind directional distributions of surface partial columns have been removed as these were only used in very few sentences of the former manuscript version. Statistical values of surface partial columns which were given along with the scatter density plots in the former manuscript version are now summarized in Table 4 (see below).

-Subfigures showing means over different seasons of vertical profiles, seasonal cycles, diurnal cycles and weekly cycles were moved to the Appendix.

As less subimages are now shown in the main part of the revised manuscript version, this freed up space for remaining ones which are now larger in size and it should now be easier for the reader to concentrate on details. Note also, that the quality of all Figures is good enough to allow zooming into them. This is especially helpful for the Figures in the Appendix containing further results from individual model runs and for different seasons.

In the previous manuscript version, standard deviations calculated based on results from individual ensemble members were used as an indicator of how much individual ensemble members differ from each other and shown along with vertical profiles as well as seasonal, diurnal and weekly cycle Figures (Figure 4, 7, 8 ,9, 10, 11 of the previous manuscript version). In the revised version, standard deviations have been removed from text and Figures which now show individual model runs in addition to the ensemble median instead (see Figure 5, 8, 9, 10, 11 of revised version). Moreover (in response to comments by reviewer #1), three Tables have been added to the main part of the manuscript (further increasing visibility of individual model results in the main part of the manuscript):

-Table 3 shows statistical values of AVK-weighted tropospheric $NO_2$ VCDs for the four stations for the ensemble and individual model runs

-Table 4 shows the same as Table 3, but for surface partial columns of $NO_2$

-Table 5 shows the same as Table 3, but for seasonal, diurnal and weekly cycles of AVK-weighted tropospheric $NO_2$ VCDs

More text on individual model results has been added in several parts of the manuscript, which also points at differences among ensemble members including:

[revised manuscript text omitted]

Note also that the abstract has been reformulated in order to reflect the performance of individual models in general.

As results of individual models were moved to the main part of the manuscript, the wording has been changed in some parts of the manuscript in order to be able to differentiate if it is referred to the ensemble or individual model results. Moreover, as standard deviations have been removed in the revised version, it is now referred to "the spread between individual models" instead, e.g.:

-(p 13 l 21-22, revised version) "In the present study, the spread between individual models is quite large for OHP indicating that some of the models perform better than others."

Regarding the use of the model ensemble median, the following text has been added in the revised version (p 9 l 21-30, revised version): "While the calculation of an ensemble median is a common approach to reduce individual model outliers, it is mainly used here for the sake of simplicity and presentation purposes, allowing easier overall evaluation of how the models compare to MAX-DOAS retrievals. The model ensemble is based on five of the seven models (though with partly different set-ups) which constitute the CAMS regional model ensemble (http://www.regional.atmosphere.copernicus.eu/) for which Marécal et al. (2015) have shown that at least for ozone, the ensemble median performs on average best in terms of statistical indicators compared to the seven individual models and that the ensemble is also robust against reducing the ensemble size by one member. Statistical indicators for $NO_2$ (see Table 3 to 5) show that the ensemble median of the present study performs best in terms of overall correlation to individual MAX-DOAS measurements at each station. Compared to individual models for other statistical indicators and also comparisons for seasonal, diurnal and weekly cycles, reasonable results are achieved by the ensemble median."

*What I find contradicting a bit in this paper is also the fact that data are used to validate an ensemble, use nature is obscure, and the main message that this brought forward is indirectly that this exercise demonstrates that Max-Doas data are suitable to validate models. So what is validating what and how?*

In the revised version, the corresponding text stating that that this study focuses on evaluating the usefulness of using MAX-DOAS data to improve model performance has been deleted (p 3 l 29-30 former version), as it was partly misleading. Moreover, the term 'validation' has been removed as suggested above. MAX-DOAS retrievals do not constitute direct measurements of $NO_2$ conditions but base on measurements of light intensity in specific wavelength windows. In this sense, they are closer to $NO_2$ conditions than simulations. This should be accounted for by a conservative overall uncertainty of MAX-DOAS retrievals of 30 % which is assumed for all stations within this manuscript and given along with the data plots, where appropriate (p 9 l 5-8 of former version, p 9 l 31- p 10 l 3 of revised version).

*In the present status the manuscript can not, in my view be published in ACP. GMD would be more suitable, but provided that more insight is given into the ensemble workings.*

This work was initially submitted to GMD, where it was regarded as out of the journal's scope with prompt recommendation to submit to ACP instead. We believe that results of the present MAX-DOAS based comparison study and differences found between simulations and retrievals are of interest to both modelling and measurement community (therefore fit to the scope of ACP) and hope that this work stimulates future studies on improving model performance.

---

## Author Comment (AC2) · 6 Sep 2018

**Response to anonymous referee #1:**

**We thank referee #1 for constructive and helpful review comments, to which we hope to have responded appropriately. A list of comments including our response is given below.**

*The paper presents a comparison of time series of tropospheric NO2 VCDs derived from 4 European MAX-DOAS stations to an ensemble of 5 regional models. The horizontal and vertical resolution of MAX-DOAS observations fits in general well to those of the regional models. Thus such a comparison is well suited to evaluate the performance of the model simulations (and also the quality of the MAX-DOAS retrievals). In this respect, the results of this paper are of high importance, and are well suited for publication in ACP. However, I have three major concerns with respect to the evaluation and presentation of the results in the present version of the manuscript, which should be addressed before final publication:*

*a) One of the main advantages of MAX-DOAS observations is that profile information for the lowest layers of the atmosphere (below about 2km) can be obtained. Profile information is crucial to assess the performance of the model simulations (and to understand deviations from observations). It is a pity (and completely unclear to me), why the authors do not make explicit use of the profile information derived from MAX-DOAS. One – rather simple – way to make use of the profile information (and to compare MAX-DOAS results and model simulations) would be to determine a characteristic layer height (e.g. the layer, below 70% of the total tropospheric column resides) from both the MAX-DOAS observations and the model results.*

In the manuscript, vertical information from MAX-DOAS is made use of by comparing average vertical profiles of simulations and retrievals (Figure 5 and A1 of revised manuscript) and described in the results section (p 11 l 9-18, revised version), demonstrating principle agreement between measured and retrieved profiles. We agree that comparisons of characteristic layer heights may show useful additional information on the ability of the models to reproduce the distribution of $NO_2$ in the vertical. However, also keeping in mind the number of Figures shown in the manuscript, we consider this as an interesting topic for future studies. The latter has been added to the summary and conclusions section on p 18 l 9-11 (revised version):

"Moreover, one could investigate the ability of the models to distribute $NO_2$ in the vertical in terms of characteristic layer height of $NO_2$, which is (in addition to other factors like vertical distribution of emissions or boundary layer schemes) expected to be affected by vertical resolution of the models."

*b) The authors compare the MAX-DOAS results to model ensembles. Although in the appendix, also the comparison results to the individual models are shown, no attempt is made to systematically asses the performance of the individual models with respect to the MAX-DOAS results. The authors should at least provide a table with some key indicators (e.g. correlation coefficient, slope, bias, etc.) for the individual model comparisons. These indicators should be provided for a) the complete time series, b) for the seasonal variation, c) the diurnal variation, and d) the weekly cycle.*

A couple of changes have been applied to the text, Figures and Tables of the manuscript in order to put more weight on results of individual models in the main part of the manuscript (this was also asked for by reviewer #2). In the revised version, three Tables have been added:

-Table 3 shows statistical values of AVK-weighted tropospheric $NO_2$ VCDs for the four stations for the ensemble and individual model runs

-Table 4 shows the same as Table 3, but for surface partial columns of $NO_2$

-Table 5 shows the same as Table 3, but for seasonal, diurnal and weekly cycles of AVK-weighted tropospheric $NO_2$ VCDs

More text on individual model results has been added in several parts of the manuscript, which also points at differences among ensemble members including:

[revised manuscript text omitted]

Note also that the abstract has been reformulated in order to reflect the performance of individual models in general.

In the previous manuscript version, standard deviations calculated based on results from individual ensemble members were used as an indicator of how much individual ensemble members differ from each other and shown along with vertical profiles as well as seasonal, diurnal and weekly cycle Figures (Figure 4, 7, 8 ,9, 10, 11 of the previous manuscript version). In the revised version, standard deviations have been removed from text and Figures which now show individual model runs in addition to the ensemble median instead (see Figure 5, 8, 9, 10, 11 of revised version).

Note also that the number of Figures and subimages has been reduced in the new version, which is both a consequence of the new Tables added and the request by reviewer #2 to increase size of the Figures:

-Figures showing non AVK-weighted tropospheric $NO_2$ VCDs (termed tropospheric $NO_2$ VCDs from method 1 in previous version) were deleted as these do not differ substantially from AVK-weighted (referred to as method 2 in previous version) values (see p 11 l 19 - p 12 l 2, revised version).

-Scatter density plots and wind directional distributions of surface partial columns have been removed as these were only used in very few sentences of the former manuscript version. Statistical values of surface partial columns which were given along with the scatter density plots in the former manuscript version are now summarized in Table 4 (see below).

-Subfigures showing means over different seasons of vertical profiles, seasonal cycles, diurnal cycles and weekly cycles were moved to the Appendix.

*c) The discussion of the deviations between the model simulations and the MAX-DOAS results is weak, and only rather general explanations for the disagreements are given. The paper would benefit a lot if the possible reasons for disagreement would be investigated in more depth. In particular, from the two points mentioned above, useful information could be obtained, which processes (e.g. transport, emission inventories, chemistry) might be most important reason for discrepancies for individual situations and/or model*

As described in reply to point b) above, the revised manuscript contains Tables showing overall statistical values for the ensemble and individual model runs and corresponding ones for seasonal, diurnal and weekly cycles. Based on the new Tables and also as part of the response to referee #2, the contribution of seasonal, diurnal and weekly cycles to overall correlations has been investigated. This showed that overall correlations reached at all stations are mainly driven by seasonal and weekly cycles, while significantly lower and in many cases negative correlations are achieved for diurnal cycles which decreases overall correlations. An exception for the latter is Uccle, where good correlations are also found for diurnal cycles. This is now described on p 15 l 22-24 of the revised version.

Moreover, diurnal cycles based on weekdays and based on weekends only have been derived and are now presented and discussed in the revised version (see p 14 l 27 – p 15 l 10, p 16 l 20-27) and a corresponding Figure showing diurnal cycles for weekends only has been added (Figure 10, revised version). Note that results for weekdays only look similar to results based on all days of the week and are therefore not shown in the manuscript. Diurnal cycles based on weekends only in general show a rather flat shape for the urban stations. However, the shape of model simulated diurnal cycles looks very similar for weekdays compared to weekends, meaning that simulations fail to reproduce the observed changes towards the weekend. It should be checked in future studies if switching off diurnal scalings of emissions during weekends leads to an improvement in model performance compared to MAX-DOAS. A note on these results has also been added to the Abstract (p 1 l 14 – p 2 l 2, revised version).

In addition to the MAX-DOAS comparisons shown in the present study, we also carried out a comparison between the regional models and OMI satellite retrievals with similar results as Huijnen et al. (2010). A paragraph on these comparisons has been added on p 17 l 1-13 of the revised version. However, due to the generally short lifetime of $NO_2$, to properly relate uncertainties in the simulations over emission hotspots indicated by the OMI based comparisons to the ones derived from MAX-DOAS based comparisons would generally require investigating transport patterns of individual model runs with much higher time resolution around the MAX-DOAS sites, which is not provided by the satellite data (only one OMI orbit per day over the stations).

A Figure showing a map of OMI satellite observations and TNO/MACC-II anthropogenic NOx emissions has also been added to the manuscript (Figure 1 in revised version, corresponding text added on p 4 l 1-4). The spatial distribution of NOx emissions agrees well with pollution hotpots and cleaner areas identified by OMI. The latter shows that the spatial distribution of emissions does not seem to be a likely reason for differences between simulations and MAX-DOAS retrievals.

The impact of horizontal model resolution on the ability of the models to reproduce MAX-DOAS results is now discussed in the revised version (p 17 l 19 - p 18 l 9). One would expect that this ability increases with increasing model resolution. However, no clear relation between model resolution and performance of the models resulted from these investigations, which shows that other differences between the models such as chemistry schemes and treatment of emissions strongly impact on comparison results. (see also reply to minor point on model resolution below)

Additional comparison results described above pointed at more likely (and also less likely) reasons for differences between simulations and observations and hence provided further useful information for future studies to track down reasons of disagreement with the aim to achieve a better agreement between MAX-DOAS and model results. This would mainly involve running models with different model set-ups, emission inventories, resolution, parameterisations and chemistry schemes. The summary and conclusions section has been extended by the results described above and more ideas for future studies are now given.

Huijnen, V., Eskes, H. J., Poupkou, A., Elbern, H., Boersma, K. F., Foret, G., Sofiev, M., Valdebenito, A., Flemming, J., Stein, O., Gross, A., Robertson, L., D'Isidoro, M., Kioutsioukis, I., Friese, E., Amstrup, B., Bergstrom, R., Strunk, A., Vira, J., Zyryanov, D., Maurizi, A., Melas, D., Peuch, V.-H., and Zerefos, C.: Comparison of OMI $NO_2$ tropospheric columns with an ensemble of global and European regional air quality models, Atmos. Chem. Phys., 10, 3273-3296, doi:10.5194/acp-10-3273-2010, 2010.

*Minor points:*

*Page 1, line 1: Replace NO2 by NOx*

Changed to: "Tropospheric NOx ($NO+NO_2$) is hazardous to human health and can lead to tropospheric ozone formation, eutrophication of ecosystems and acid rain production."

*Page 1, line 8: 'measurements are available during daylight'. To me it seems that this is not an advantage but rather a disadvantage (measurements are not available during night)*

Thanks for pointing this out. More explicitly, the advantage the sentence should have referred to is, that multiple measurements are carried out during daylight, so that e.g. diurnal cycles can be derived from the retrievals. The sentence has been changed to (p 1 l 6-9, revised version):

"Compared to other observational data usually applied for regional model evaluation, MAX-DOAS data is closer to the regional model data in terms of horizontal and vertical resolution and multiple measurements are available during daylight, so that for example diurnal cycles of trace gases can be investigated."

*Introduction: It should be made more clear, that the quantity of interest is NOx, but only NO2 can be measured*

Added the following sentence (p 3 l 21-22, revised version):

"In contrast to $NO_2$, NOx cannot be retrieved from MAX-DOAS measurements directly, so that these measurements are of more interest for air quality than for atmospheric chemistry studies."

*Page 2, line 30: The statement 'using zenith measurements as intensity of incident radiation' is unclear to me. Do you mean incident solar irradiation? Then I would disagree. Please clarify.*

This sentence was misleading and has been rephrased to (p 3 l 1-3, revised version):

" Therefore, using observations in low elevation angles as measurement intensity and zenith measurements as reference intensity, the total amount of molecules of a certain species along the light

path difference (zenith subtracted from non-zenith measurement), so called differential slant column densities, can be determined using Lambert Beer's law."

*Section 2.1: What is the spatial resolution of the models? How does it compare to the horizontal sensitivity ranges of the MAX-DOAS results?*

In response to this question, the following text has been added to p 17 l 19 - p 18 l 9 of the revised manuscript (this is combined with a response to referee #2 who also asked about the impact of model resolution on comparison results):

"The horizontal grid spacing (Table 1) differs for the 6 model runs evaluated in the present study, with a resolution of approximately 9x7 $km^2$ for the highest resolution run (LOTOS-EUROS) and 50x50 $km^2$ for the coarsest one (EMEP). The resolution of the remaining model runs is approximately 20x20 $km^2$. As described in Section 2.2, the horizontal averaging volume of MAX-DOAS retrievals strongly depends on aerosol loading, viewing direction and wavelength (Richter et al., 2013). As a rough estimate, it ranges from 5 to 10 km for the stations used in the present study. Therefore, the horizontal averaging volume is (apart from the coarsest resolution run) expected to be either on the same spatial scale as the horizontal model resolution or by a factor of 1 to 4 smaller. From the latter (i.e. horizontal averaging volume of MAX-DOAS smaller than model resolution) one would expect an underestimation of enhancements in tropospheric columns observed by MAX-DOAS in case of horizontal changes in tropospheric $NO_2$ columns below the model resolution and, similarly, an overestimation of local minima in tropospheric $NO_2$ columns. However, in reality, the comparison between horizontal averaging volume of MAX-DOAS and horizontal resolution of the models is much more complicated, as MAX-DOAS instruments usually measure in one azimuthal pointing direction meaning that measurements are performed only on a specific line of sight whereas model simulations are performed for three dimensional grid boxes. This could for example mean that a pollution plume with a horizontal extent on the order of the model resolution and hence showing up in the simulations is missed by the line of sight of the MAX-DOAS instrument. It would therefore be desirable to perform multiple MAX-DOAS measurements over a range of different azimuthal angles for each station and use these in future model to MAX-DOAS comparison studies.

A pollution plume and related increase in the time series of tropospheric $NO_2$ VCDs observed by MAX-DOAS would be expected to be reproduced better by model runs with higher horizontal resolution compared to lower resolution runs. The lifetime of $NO_2$ is also expected to increase with model resolution. However, in the present study, the LOTOS-EUROS run with significantly higher horizontal resolution than the other runs in general did not perform better than lower resolution runs which can probably be explained by its low number of vertical layers. Similarly, the EMEP run with significantly lower horizontal resolution did not perform worse than higher resolution runs, which shows that other differences between the models such as chemistry schemes and treatment of emissions strongly impact on comparison results. It would be interesting to investigate the ability of the models to predict the scales of $NO_2$ spatial variations derived from time scales of $NO_2$ variations and wind speeds in the context of model resolution in a future study. "

Richter, A., Godin, S., Gomez, L., Hendrick, F., Hocke, K., Langerock, B., van Roozendael, M., Wagner, T.: Spatial Representativeness of NORS observations, NORS project deliverable, available online at: http://nors.aeronomie.be/projectdir/PDF/D4.4_NORS_SR.pdf, 2013.

*Section 2.2: The retrievals are described in an inconsistent and partly incomplete way. For example, for KNMI the retrieval procedure is completely unclear. Was a profile inversion performed or not? This section should be harmonised and completed. The effect of the different inversion procedures on the NO2 results should be briefly discussed.*

This section has been harmonized. In the first paragraph, a brief general description of how $NO_2$ profiles/columns are derived from the measurements is given. For each station, the most important retrieval and measurement site information are then given (such as instrument type, location and pointing direction of instrument, wavelength window of instrument and of the $NO_2$ DOAS fit, the radiative transfer model used, cross sections of gases included in the fit, how a-priori profiles were derived). Moreover, the retrieval procedure for De Bilt is now described in more detail.

*Section 2.2: It is stated that for Uccle, cloud information was retrieved. Was this information also used for the selection of the measurements? What about the retrieval of cloud information for the other stations?*

The following text is now given in the last paragraph of Section 2.2 (p 7 l 22-27, revised version):

"For Uccle, information on cloud conditions was retrieved according to the method by Gielen et al. (2014) which is based on analysis of the MAX-DOAS retrievals, but not applied for results shown in the present study. No cloud flags are available for Bremen, De Bilt and OHP. Larger uncertainties are associated with retrievals under cloudy conditions in particular as clouds are not included in the MAX-DOAS forward calculations. However, MAX-DOAS retrievals are usually filtered for patchy cloud situations by comparing radiative forward calculations of $O_4$ to retrieved $O_4$ columns and removing cases from the data with larger than expected differences."

Note that the discussion and analysis of the impact of clouds on comparison results has been removed from the results section (as suggested by anonymous referee #2) and regarded as a topic for future studies, which is now mentioned on p 7 l 34 and p 18 l 21 of the revised manuscript.

Gielen, C., Van Roozendael, M., Hendrick, F., Pinardi, G., Vlemmix, T., De Bock, V., De Backer, H., Fayt, C., Hermans, C., Gillotay, D., and Wang, P.: A simple and versatile cloud-screening method for MAX-DOAS retrievals, Atmos. Meas. Tech., 7, 3509-3527, doi:10.5194/amt-7-3509-2014, 2014.

*Section 2.3: How does the wind data compare to the wind fields used in the models?*

As described in section 2.1, all models use ECMWF-IFS as meteorological input and boundary conditions. As the models are run with differing horizontal and vertical resolution (see Table 1), wind data from the model output is expected to differ among the models. Wind speed and direction was provided as an output parameter for two of the model runs (LOTOS-EUROS and MOCAGE) of the present study. Figure R1 below shows wind directional distributions of wind speeds from the weather station data and the ones from the model output (near surface level) for the four MAX-DOAS stations (note that MOCAGE data is not available for OHP). Figure R2 shows corresponding wind directional distributions of the data percentage in each bin (e.g., a value of 10 for the 0 to 45° wind direction bin means that during 10% of the time period the wind was blowing from north to north-east). Statistical values of the wind speed comparisons were calculated along with the plots. Wind speed correlations are high for De Bilt and Bremen for both models (~0.8) and moderate for Uccle and OHP (~0.5-0.6). Wind speeds are positively biased for the three urban stations, with the largest biases for Uccle (on the order of 3 m/s), while there is a negative bias at OHP (~ -7 m/s). Note that the negative bias may result from the fact that wind speeds and directions from near surface level were taken for the comparisons which should be comparable to measurements at meteorological sites. However, this is probably not representative of winds at the small hill where the OHP station is located (~650 m above mean sea level) since the orography of the IFS model is a smoothed version of the real orography. Thus, IFS simulates wind speeds for a more flat terrain, which are therefore lower than the measured ones.

Not considering the magnitude of values, wind directional distributions of wind speed from the models agree well with the ones from the weather station data for all stations apart from Uccle. For the latter, the model output shows the highest average wind speeds to the west/south-west of the station, while the measurements show the highest ones to the north-east. As for wind speeds, wind directional distributions also agree well in general for the data percentage. Larger differences occur for Uccle for south to south-westerly and west to north-westerly wind directions and for OHP for west to north-westerly winds.

Note that wind directional distributions shown in the manuscript (Figures 7 and A3 of revised version) are (as described in the corresponding Figure captions) based on wind directions from weather station measurements solely. However, due to the generally good agreement between measured and simulated wind speeds and directions described above, this is not expected to have a strong impact on the data analysis and conclusions given in the manuscript. This is demonstrated by Figures R3 and R4 below which show wind directional distributions of tropospheric $NO_2$ VCDs for (left) LOTOS-EUROS and (right) MOCAGE based on wind directions from measurements only (as in the manuscript) as well as based on measured wind directions for MAX-DOAS retrieved values of $NO_2$ and based on model output for simulated $NO_2$ values, respectively. Overall both Figures show a good agreement between measured and simulated wind directional distributions of $NO_2$.

*What about wind data for KNMI?*

The following sentence has been added to section 2.3 (p 8 l 10-11, revised version):

"For De Bilt, wind measurements (within 300 m from the MAX-DOAS instrument) carried out by KNMI were downloaded from https://www.knmi.nl/nederland-nu/klimatologie/uurgegevens."

*Page 8, line 22: 'Only those model values closest to the measurement time are used'. Why is no interpolation in time of neighbouring model output values performed?*

This was mainly done to save computation time. As the time difference between simulations and retrievals is shorter than half an hour, interpolation in time is not expected to have a major impact on conclusions of this study.

*Page 9, line 10: What is the vertical extension of the lowest measurement layer?*

Bremen 50 m, De Bilt 180 m, Uccle 180 m, OHP 150 m above ground. This has been added to p 10 l 29-30 of the revised version.

*Page 9, line 12: 'comparisons of profiles'? No comparison of profiles is shown in Figs. 1 and 2.*

This was done in order to explain why surface partial columns are not shown in Figure 2 of previous version (Figure 3 of revised version) for De Bilt. Surface partial columns have been derived for

stations with vertical profile retrievals only. The sentence was however misleading and has been replaced by the following text in the revised version (p 10 l 28-31):

"In the present study, surface partial columns refer to the partial column of the lowest measurement layer (Bremen 50 m, De Bilt 180 m, Uccle 180 m, OHP 150 m above ground). As vertical profiles are not available from the MAX-DOAS output for De Bilt, comparisons of surface partial columns are not given for this station in the present manuscript."

*Page 10, line 5: 'As the sensitivity of MAX-DOAS retrievals is largest in the boundary layer' Is this also true for the 'de Bilt measurements'?*

Yes, the sensitivity to $NO_2$ in the boundary layer is intrinsic for the measurement method. Differences in retrieval methods will not change this. The corresponding sentence has been changed to (p 11 l 19-21, revised version) :

"As the sensitivity of MAX-DOAS retrievals is largest in the boundary layer, a feature which is independent of the retrieval method, we initially expected the application of column AVKs from the measurements to model simulations to be of crucial importance for evaluation results."

*Page 10, lines 23,24: 'On average, observed NO2 partial columns are higher in the lowest observation layers during cloudy conditions compared to clear-sky conditions' I guess that no clouds are considered in the MAX-DOAS forward model. How reliable are then the MAX-DOAS NO2 results under cloudy conditions?*

As described above, the discussion and analysis of the impact of clouds on comparison results has been removed from the results section (as suggested by anonymous referee #2) and regarded as a topic for future studies (see p 7 l 34 and p 18 l 21of revised manuscript).

Larger uncertainties are associated with retrievals under cloudy conditions in particular as clouds are not included in the MAX-DOAS forward calculations. However, MAX-DOAS retrievals are usually filtered for patchy cloud situations by comparing radiative forward calculations of $O_4$ to retrieved $O_4$ columns and removing cases from the data with larger than expected differences. This is now mentioned on p 7 l 24-27 of the revised manuscript.

*Page 11, line 3: What is exactly meant with 'correlation'? r or r squared?*

Correlations calculated in this study refer to the pearson correlation coefficient, i.e. r not squared. The latter was mentioned in the caption of Figure 5 only of the previous manuscript version, but is now mentioned in several parts of the revised manuscript (i.e. p 11 l 24, p 12 l 5, caption of Figure 6, caption of Figure A2, caption of Table 3).

*Page 11, line 12: How consistent are the wind data from the weather stations with the wind fields used in the models? Can you show a similar plot as Fig. 6 based on the wind fields from the models?*

See response to comment on section 2.3 above and corresponding Figures below. Note that this sentence has been changed to (p 12 l 26-28, revised version):

"Figure 7 shows comparisons between MAX-DOAS and the model ensemble of wind directional distributions of average tropospheric $NO_2$ VCDs based on wind measurements from station data

(note that further analysis has shown a good agreement between measured wind speeds and wind directions and those of the simulations). "

*Page 13, line 15: 'However, many validation points arise from the MAX-DOAS based comparisons which could improve model performance substantially.' This sentence is not clear to me. Please clarify.*

Although there is good agreement between MAX-DOAS retrievals and model simulations of tropospheric $NO_2$ in a general sense, differences have been found for example for individual pollution plumes observed by MAX-DOAS, seasonal, weekly and diurnal cycles. The reasons for the differences should be identified in future studies and several aspects of simulations could be changed in order to achieve a better agreement to MAX-DOAS retrievals. The corresponding sentence has been changed, we hope it is now more clear (p 16 l 2-4, revised version):

"However, many points to evaluate arise from the MAX-DOAS based comparisons. Tracking down the reasons for differences between simulations and retrievals and adjusting model runs accordingly (in case of differences caused by errors in simulations rather than uncertainties of the retrievals) could improve model performance substantially."

Text on how a better agreement to MAX-DOAS (where desirable) could be achieved has been added to section 5 (p 18 l 22-30, revised version):

"To track down reasons for the reported uncertainties of regional model simulations constitutes the main challenge for future studies. This could be achieved by running models with different chemistry schemes combined with different resolutions where possible (uncertainties in chemistry such as lifetime of $NO_2$), running models with and without scaling of emissions in time and for specific seasons or days only (uncertainties in seasonal, diurnal and weekly cycles related to emissions), performing runs with varying vertical scalings of emissions (uncertainties in injection heights) and carrying out runs with varying boundary layer physics (uncertainties of $NO_2$ profiles due to mixing of emissions in the boundary layer and transport therein). Especially LOTOS-EUROS and MOCAGE showed large differences to the MAX-DOAS retrieved seasonal and diurnal cycles for Bremen and De Bilt and also EMEP-MACCEVA for Bremen, so that the impact of different set-ups in emissions and chemistry is expected to be more pronounced compared to the other models at these stations."

[Figure]

*Figure R1: Average wind speed in 45° wide wind direction bins from (blue solid lines) weather station measurements and (red dashed lines) model output for (left) LOTOS-EUROS and (right) MOCAGE for (first row) De Bilt, (second row) Bremen, (third row) Uccle and (bottom row) OHP. Wind directions correspond to the direction towards the station and are taken from weather station measurements itself for measured and from model output for simulated wind speeds. The printed numbers in each bin refer to the number of data values used for calculating average values for each bin.*

[Figure]

*Figure R2: As in Figure R1 but for average percentage of data values. The printed numbers given for each bin were rounded to its closest integer value.*

[Figure]

*Figure R3: As in Figure R1 but for average AVK-weighted tropospheric NO₂ VCDs [10¹⁵ molec cm⁻²]. Wind directions correspond to the direction towards the station and are taken from weather station measurements for both MAX-DOAS retrieved and model simulated values.*

[Figure]

Figure R4: As in Figure R1 but for average AVK-weighted
tropospheric NO₂ VCDs [10¹⁵ molec cm⁻²]. Wind directions
correspond to the direction towards the station and are taken from
weather station measurements for MAX-DOAS retrieved and from
model output for simulated values.

---

## Author Comment (AC3) · 6 Sep 2018

**Response to anonymous referee #2:**

**We thank referee #2 for constructive and helpful review comments, to which we hope to have responded appropriately. A list of comments including our response is given below.**

*In "Comparison of tropospheric NO2 columns from MAX-DOAS retrievals and regional air quality model simulations," the authors provide a nice overview of 1) long-term MAX-DOAS records of NO2 in northwest and southwest Europe, 2) a description of regional air quality models used in the CAMS ensemble, and 3) a description of past comparisons of regional CTMs and MAX-DOAS with in situ and satellite data. The comparison of the model ensemble and the four MAX-DOAS NO2 datasets showed general agreement in a broad sense. The authors highlight when and where there are discrepancies between ensemble median model results and MAX-DOAS observations (e.g., seasonal cycle, diurnal cycle), but do not offer ideas on potential approaches for disentangling the causes of these discrepancies.*

*I felt that the paper lacked a final synthesis, written in more general language, of how future simulations and MAX-DOAS deployments like these can isolate effects from individual processes. I hope that the authors consider adding a broader synthesis of their results to the end of section 4, offering possible paths forward for future analyses: what common and distinct attributes of these four sites share? How might these differences and similarities be exploited to investigate chemistry? Emissions? Meteorology? Where might the authors propose future MAX-DOAS instruments be located? Should one expect an ensemble median to capture hourly NO2 variations? Monthly averages? What is the native scale of NO2 spatial variations at the MAX-DOAS sites inferred from the time scale of NO2 variation and wind speed?*

Many changes have been applied to the summary and conclusions section in the revised version including for example a discussion of model resolution and averaging volume of MAX-DOAS measurements, suggestions for sites to investigate in future studies (i.e. stations affected by different meteorological and pollution conditions for example at pollution hotspots in the Mediterranean with strong smog conditions especially during summer and clean mountain sites), a paragraph on OMI satellite comparisons with similar results as in Huijnen et al. (2010), as well as further suggestions on how to track down reasons for differences between model runs and MAX-DOAS retrievals (please see Section 5 of revised manuscript for further details).

Huijnen, V., Eskes, H. J., Poupkou, A., Elbern, H., Boersma, K. F., Foret, G., Sofiev, M., Valdebenito, A., Flemming, J., Stein, O., Gross, A., Robertson, L., D'Isidoro, M., Kioutsioukis, I., Friese, E., Amstrup, B., Bergstrom, R., Strunk, A., Vira, J., Zyryanov, D., Maurizi, A., Melas, D., Peuch, V.-H., and Zerefos, C.: Comparison of OMI $NO_2$ tropospheric columns with an ensemble of global and European regional air quality models, Atmos. Chem. Phys., 10, 3273-3296, doi:10.5194/acp-10-3273-2010, 2010.

*Comments: P2, L10-11: NO2 lifetime is much longer in the upper troposphere, primarily because its chemical family, NOx, is mostly present as NO at high altitudes, which has far fewer permanent sinks.*

Changed to (p 2 l 14-16): "The lifetime of NOx is only a few hours in the boundary layer but a few days in the upper troposphere, where less OH radicals are present (Ehhalt et al., 1992) to react with $NO_2$ and more NOx is present as NO which has fewer permanent sinks than $NO_2$."

*P3, L29: "focusses" – typo*

This sentence has been deleted in response to a comment by referee #3.

*P4: There is no discussion of model resolution. The NO2 lifetime is a function of model resolution. Also, median values may be biased towards coarser models as those with finer resolution may produce highs when a plume passes and lows when not.*

In response to this question, the following text has been added to p 17 l 19 - p 18 l 9 of the revised manuscript (as response to a comment by referee #1, this is combined with a description on how the horizontal sensitivity range of MAX-DOAS compares to model resolution):

"The horizontal grid spacing (Table 1) differs for the 6 model runs evaluated in the present study, with a resolution of approximately 9x7 $km^2$ for the highest resolution run (LOTOS-EUROS) and 50x50 $km^2$ for the coarsest one (EMEP). The resolution of the remaining model runs is approximately 20x20 $km^2$. As described in Section 2.2, the horizontal averaging volume of MAX-DOAS retrievals strongly depends on aerosol loading, viewing direction and wavelength (Richter et al., 2013). As a rough estimate, it ranges from 5 to 10 km for the stations used in the present study. Therefore, the horizontal averaging volume is (apart from the coarsest resolution run) expected to be either on the same spatial scale as the horizontal model resolution or by a factor of 1 to 4 smaller. From the latter (i.e. horizontal averaging volume of MAX-DOAS smaller than model resolution) one would expect an underestimation of enhancements in tropospheric columns observed by MAX-DOAS in case of horizontal changes in tropospheric $NO_2$ columns below the model resolution and, similarly, an overestimation of local minima in tropospheric $NO_2$ columns. However, in reality, the comparison between horizontal averaging volume of MAX-DOAS and horizontal resolution of the models is much more complicated, as MAX-DOAS instruments usually measure in one azimuthal pointing direction meaning that measurements are performed only on a specific line of sight whereas model simulations are performed for three dimensional grid boxes. This could for example mean that a pollution plume with a horizontal extent on the order of the model resolution and hence showing up in the simulations is missed by the line of sight of the MAX-DOAS instrument. It would therefore be desirable to perform multiple MAX-DOAS measurements over a range of different azimuthal angles for each station and use these in future model to MAX-DOAS comparison studies.

A pollution plume and related increase in the time series of tropospheric $NO_2$ VCDs observed by MAX-DOAS would be expected to be reproduced better by model runs with higher horizontal resolution compared to lower resolution runs. The lifetime of $NO_2$ is also expected to increase with model resolution. However, in the present study, the LOTOS-EUROS run with significantly higher horizontal resolution than the other runs in general did not perform better than lower resolution runs which can probably be explained by its low number of vertical layers. Similarly, the EMEP run with significantly lower horizontal resolution did not perform worse than higher resolution runs, which as expected shows that other differences between the models such as chemistry schemes and treatment of emissions strongly impact on comparison results. It would be interesting to investigate the ability of the models to predict the scales of $NO_2$ spatial variations derived from time scales of $NO_2$ variations and wind speeds in the context of model resolution in a future study."

Richter, A., Godin, S., Gomez, L., Hendrick, F., Hocke, K., Langerock, B., van Roozendael, M., Wagner, T.: Spatial Representativeness of NORS observations, NORS project deliverable, available online at: http://nors.aeronomie.be/projectdir/PDF/D4.4_NORS_SR.pdf, 2013.

*P5, L7: These sites, with exception of OHP, seem to be in very similar physical settings, with likely similar meteorology (e.g., vertical mixing characteristics). If so, this fact should be mentioned.*

This is now mentioned in the summary and conclusions section together with suggestions for MAX-DOAS sites to be incorporated in future comparison studies (p 18 l 16-19):

"As the stations investigated in the present study have, apart from the rural background station OHP, rather similar meteorological and pollution conditions, investigation of stations over a broader range of different conditions would be desirable. Further comparison studies could for instance include stations at pollution hotspots in the Mediterranean such as Athens with strong smog conditions especially during summer and clean mountain sites."

*Please also consider including a map of the region with sites indicated on a backdrop of satellite-based tropospheric NO2 column measurements.*

The location of the MAX-DOAS stations is now shown in Figure 1 of the revised version, plotted on top of mean tropospheric columns of $NO_2$ from OMI for February 2011 as well as on top of TNO/MACC-II anthropogenic NOx emissions as an indicator of pollution levels in these and surrounding regions. The spatial distribution of NOx emissions agrees well with pollution hotpots and cleaner areas identified by OMI. Corresponding text has been added on p 4 l 1-4 of the revised version. The latter shows that the spatial distribution of emissions does not seem to be a likely reason for differences between simulations and MAX-DOAS retrievals.

*Minor comment: I did not see Lat/Lon values reported for Uccle.*

Added to revised version on p 6 l 33

*Page 6, Line 29: Has there been any side-by-side operation and comparison of these two instruments? If so, please provide the reference.*

The Uccle and OHP MAXDOAS instruments are a commercial mini-MAX-DOAS from Hoffmann Messtechnik GmbH and a BIRA research-grade spectrometer, respectively. Although there has not been formal side-by-side operation of both instruments for verification purpose, a good overall agreement has been obtained between the mini-DOAS and other BIRA research-grade spectrometers similar to the one operated at OHP, e.g. like during the CINDI campaign (see Roscoe et al., 2010). The last sentence has been added to p 7 l 14-17 of the revised manuscript.

Roscoe, H. K., Van Roozendael, M., Fayt, C., du Piesanie, A., Abuhassan, N., Adams, C., Akrami, M., Cede, A., Chong, J., Clémer, K., Frieß, U., Gil Ojeda, M., Goutail, F., Graves, R., Griesfeller, A., Grossmann, K., Hemerijckx, G., Hendrick, F., Herman, J., Hermans, C., Irie, H., Johnston, P. V., Kanaya, Y., Kreher, K., Leigh, R., Merlaud, A., Mount, G. H., Navarro, M., Oetjen, H., Pazmino, A., Perez-Camacho, M., Peters, E., Pinardi, G., Puentedura, O., Richter, A., Schönhardt, A., Shaiganfar, R., Spinei, E., Strong, K., Takashima, H., Vlemmix, T., Vrekoussis, M., Wagner, T., Wittrock, F., Yela, M., Yilmaz, S., Boersma, F., Hains, J., Kroon, M., Piters, A., and Kim, Y. J.: Intercomparison of slant column measurements of $NO_2$ and $O_4$ by MAX-DOAS and zenith-sky UV and visible spectrometers, Atmos. Meas. Tech., 3, 1629–1646, doi:10.5194/amt-3-1629-2010, 2010.

*P9, L5: "As the typical error on MAX-DOAS retrieved VCDs is around 20%" – please describe this statement in more detail: at what time scale? Random or systematic uncertainty? Based on measurement intercomparisons or fitting statistics?*

Uncertainty discussion of MAX-DOAS measurements is complex but has been done in previous studies (e.g. Hendrick et al., 2014; Wang et al., 2014; Franco et al., 2015). Briefly, uncertainties are a combination of small systematic errors (for example from the cross-sections used), random errors resulting from the DOAS retrieval, errors introduced by the profile retrieval and a priori assumptions made. In particular the latter contribution can vary depending on aerosol loading, vertical $NO_2$ profile and cloud contamination. In polluted conditions, uncertainties from profiling dominate. In clean situations, random errors from the fit can become significant. In general, uncertainties can be considered as random or pseudo-random, but systematic errors can result from, for example, the presence of elevated aerosol layers.

Quantification of uncertainties not only from error propagation but also from validation with independent measurements would be desirable, but very few suitable validation measurements are available, and differences are usually dominated by differences in measurement volume. Intercomparisons of different DOAS instruments show excellent (a few percent deviations) agreement on the level of slant columns (e.g. Roscoe et al., 2010) but substantial (20% - 50%) differences at the level of profiles.

Here, a simplified and conservative estimate of 30% uncertainty on all MAX-DOAS measurements has been assumed. Data products with more detailed uncertainty information are currently in development for example in the framework of the FRM4DOAS project (http://frm4doas.aeronomie.be/), and once available, this data and related uncertainty information should be used in future comparison studies.

The last sentence of the previous paragraph has been added on p 10 l 1-3 of the revised version.

Franco, B., Hendrick, F., Van Roozendael, M., Müller, J.-F., Stavrakou, T., Marais, E. A., Bovy, B., Bader, W., Fayt, C., Hermans, C., Lejeune, B., Pinardi, G., Servais, C., and Mahieu, E.: Retrievals of formaldehyde from ground-based FTIR and MAX-DOAS observations at the Jungfraujoch station and comparisons with GEOS-Chem and IMAGES model simulations, Atmos. Meas. Tech., 8, 1733-1756, doi:10.5194/amt-8-1733-2015, 2015.

Hendrick, F., Müller, J.-F., Clémer, K., Wang, P., De Mazière, M., Fayt, C., Gielen, C., Hermans, C., Ma, J. Z., Pinardi, G., Stavrakou, T., Vlemmix, T., and Van Roozendael, M.: Four years of ground-based MAX-DOAS observations of HONO and $NO_2$ in the Beijing area, Atmos. Chem. Phys., 14, 765–781, doi:10.5194/acp-14-765-2014, 2014.

Roscoe, H. K., Van Roozendael, M., Fayt, C., du Piesanie, A., Abuhassan, N., Adams, C., Akrami, M., Cede, A., Chong, J., Clémer, K., Frieß, U., Gil Ojeda, M., Goutail, F., Graves, R., Griesfeller, A., Grossmann, K., Hemerijckx, G., Hendrick, F., Herman, J., Hermans, C., Irie, H., Johnston, P. V., Kanaya, Y., Kreher, K., Leigh, R., Merlaud, A., Mount, G. H., Navarro, M., Oetjen, H., Pazmino, A., Perez-Camacho, M., Peters, E., Pinardi, G., Puentedura, O., Richter, A., Schönhardt, A., Shaiganfar, R., Spinei, E., Strong, K., Takashima, H., Vlemmix, T., Vrekoussis, M., Wagner, T., Wittrock, F., Yela, M., Yilmaz, S., Boersma, F., Hains, J., Kroon, M., Piters, A., and Kim, Y. J.: Intercomparison of slant column measurements of $NO_2$ and $O_4$ by MAX-DOAS and zenith-sky UV and visible spectrometers, Atmos. Meas. Tech., 3, 1629–1646, doi:10.5194/amt-3-1629-2010, 2010.

Wang, T., Hendrick, F., Wang, P., Tang, G., Clémer, K., Yu, H., Fayt, C., Hermans, C., Gielen, C., Pinardi, G., Theys, N., Brenot, H., and Van Roozendael, M.: Evaluation of tropospheric SO2 retrieved from MAX-DOAS measurements in Xianghe, China, Atmos. Chem. Phys. Discuss., 14, 6501-6536, doi:10.5194/acpd-14-6501-2014, 2014.

*Page 9, L17-21: See comment on "page 4" above. NOx lifetime depends on model resolution, and NO2 maxima will be diluted in coarser models. Model resolution needs to be better reported.*

See reply above.

*P10, L5-17: I have a hard time following the language and reasoning behind this conclusion. Please consider clarifying. Is this because the a priori profiles are generated from similar models*

*as those included in the comparison? Are any systematic effects buried below random sources of uncertainty?*

Multiplying simulated $NO_2$ partial columns by column AVKs of the retrievals prior to summing up partial columns in the vertical does not have a big impact on derived tropospheric $NO_2$ VCDs. One of the reasons for this is that (as shown by Figure 5 and A1, revised version), AVKs are close to 1 around the boundary layer where MAX-DOAS instruments have the highest sensitivity (generally a bit larger than one close to the surface and smaller than one higher up which has a balancing effect) and that the vertical shape of the column AVK curve is in principal agreement with the shape of simulated $NO_2$ partial columns. At altitudes above roughly 1 km, AVKs are on average for some stations significantly smaller than one, but simulated $NO_2$ partial columns are also significantly smaller at these altitudes compared to lower levels, so that the contribution to the tropospheric column is limited. At higher altitudes, MAX-DOAS retrievals tend to follow the a-priori, while retrievals in the boundary layer are not much influenced by the a-priori in general. This is in contrast to the situation for satellite observations of tropospheric $NO_2$, which usually have a minimum of the AVK in the boundary layer, i.e. where the largest fraction of $NO_2$ is usually located in polluted situations. A-priori profiles used within the MAX-DOAS retrievals (see Section 2.2) are in principal agreement with the ones simulated by the models. The vertical weighting caused by application of AVKs to partial columns does therefore not significantly impact on derived tropospheric $NO_2$ VCDs.

The information given in the paragraph above has been added to the results section and the corresponding text changed accordingly (see p 11 l 19 - p 12 l 2, revised version). Note that no profile retrievals are performed at De Bilt, which is therefore not shown in Figure 5.

Information on how a-priori profiles were derived for each station has been added to section 2.2. For Uccle and OHP, exponentially decreasing a-priori profiles were constructed based on an estimation of vertical column densities derived by so-called geometrical approximation (Hönninger et al., 2004; Brinksma et al., 2008) using scaling heights of 1 km and 0.5 km, respectively. For Bremen, an a-priori profile which is constant with height has been assumed in the retrieval. For De Bilt, a-priori profiles of $NO_2$ are based on a block-profile with $NO_2$ present the boundary layer, boundary layer heights were taken from a climatology based on ECMWF data.

Brinksma, E.J., Pinardi, G. J., Braak, R., Volten, H., Richter, A., Dirksen, R. J., Vlemmix, T., Swart, D. P. J., Knap, W. H., Veefkind, J. P., Eskes, H. J., Allaart, M., Rothe, R., Piters, A. J. M., and Levelt, P.F.: The 2005 and 2006 DANDELIONS $NO_2$ and Aerosol Intercomparison Campaigns. J. Geophys. Res.,113, D16S46, doi:10.1029/2007JD008808, 2008.

Hönninger, G., von Friedeburg, C., and Platt, U.: Multi axis differential optical absorption spectroscopy (MAX-DOAS), Atmos. Chem. Phys., 4, 231-254, doi:10.5194/acp-4-231-2004, 2004.

*Page 10, L21-31: This analysis and discussion is tangential to the broader scope of the paper and should be removed, as earlier noted by the authors "The impact of clouds on MAX-DOAS retrievals is described in detail by Vlemmix et al. (2015)" I do consider the comparison of model and MAX-DOAS NO2 columns under different cloud conditions to be an interesting topic for its own manuscript.*

The discussion and analysis of the impact of clouds on comparison results has been removed from the results section as suggested and is regarded as a topic for future studies, which is now mentioned on p 7 l 34 and p 18 l 21 of the revised manuscript.

*P10-11, L34-11: How much of the correlation is determined by seasonal and weekly cycle? Consider isolating correlation at one time of day, one season and one set of weekdays (e.g., M-F)*

In response to this comment and comment b) by referee #1 three Tables have been added to the manuscript (note that in these Tables also results of individual model runs are summarized, in response to the requests by the other two referees to put more weight on individual model results in the main part of the manuscript):

-Table 3 shows statistical values of AVK-weighted tropospheric $NO_2$ VCDs for the four stations for the ensemble and individual model runs

-Table 4 shows the same as Table 3, but for surface partial columns of $NO_2$

-Table 5 shows the same as Table 3, but for seasonal, diurnal and weekly cycles of AVK-weighted tropospheric $NO_2$ VCDs

The following text has been added on p 15 l 22-24 of the revised version:

"Comparing Table 3 and 5 shows, that the overall correlations reached at all stations are mainly driven by seasonal and weekly cycles, while significantly lower and in many cases negative correlations are found for diurnal cycles which decreases overall correlations. An exception for the latter is Uccle, where good correlations are also found for diurnal cycles. "

*P12, L35: Consider a reference to Beirle et al. (2003). I think that this paragraph could be expanded. Day-of-week effects, over the long-term, are independent of meteorology and driven entirely by variations of emissions and chemistry. Future day-of-week comparisons would be one means of providing systematic approaches to quantify the many processes affecting NO2 (emissions, meteorology uncertainty, chemistry, observational uncertainty)*

A reference to Beirle et al. (2003) has been added to p 15 l 19-21 of revised version:

"Beirle et al. (2003) investigated weekly cycles of tropospheric $NO_2$ based on GOME satellite observations and found a decrease in values of up to about 50 % towards Sundays over polluted regions and cities in Europe. This is in principal agreement with results of the present study, although the choice of the cities is different."

Differences in diurnal cycles derived for weekdays and derived for weekends only are now presented and discussed in the revised version (see p 14 l 27 – p 15 l 10, p 16 l 20-27) and a corresponding Figure showing diurnal cycles for weekends only has been added (Figure 10, revised version). Note that results for weekdays only look similar to results based on all days of the week and are therefore not shown in the manuscript. As expected, diurnal cycles retrieved from MAX-DOAS based on weekends only in general show a rather flat shape for the urban stations. However, the shape of model simulated diurnal cycles looks very similar for weekdays compared to weekends, meaning that simulations fail to reproduce the observed changes towards the weekend. It should be checked in future studies if switching off diurnal scalings of emissions during weekends leads to an improvement in model performance compared to MAX-DOAS. A note on these results has also been added to the Abstract (p 1 l 14 – p 2 l 2, revised version).

---

## Author Response (AR2)

Dear Dr. Tim Butler,

Many thanks for the handling of the review process of 'Comparison of tropospheric $NO_2$ columns from MAX-DOAS retrievals and regional air quality model simulations'
(MS No.: acp-2016-1003). The response to the referee comments and a changes tracked version (deleted text in red, new text in blue font) is given on the next pages.

In particular, in response to the comments of referee #2, text which only lists possible reasons for the differences in model results and MAX-DOAS retrievals has been removed from the manuscript and the 'Summary and Conclusions' Section has been largely rewritten. The results of this study show that future model development needs to concentrate on improving representation of diurnal cycles and associated temporal scalings. This is now pointed out more clearly in the manuscprript.

Apart from these major changes, the text and wording has been further improved, references were updated and a few remaining typos were corrected.

Sincerely,

Anne-Marlene Blechschmidt

**Response to anonymous referee #1:**

**We thank referee #1 for constructive, helpful comments and for taking the time for the review. We hope to have answered the remaining comments/questions. Our response is given below.**

*The authors addressed most of my comments, and the manuscript is largely improved.*

*Before publication, I recommend to address the points listed below:*

*-Abstract:*
*The sentence 'A large number of evaluation points arise from the comparison to MAX-DOAS measurements.' is not clear to me. What exactly is meant with 'comparison points'?*

*- Summary and conclusions:*
*Also the sentence 'However, many points to evaluate arise from the MAX-DOAS based comparisons.' is not clear to me.*

We decided to remove the sentences from the manuscript as they seem to have been misleading.

*Page 20, line 5: It is stated that 'The lifetime of NO2 is also expected to increase with model resolution.'*
*Why is this expected? Can a reference be given?*

According to Valin et al. (2011), the lifetime depends non-linearly on the concentration of $NO_2$ and hence also on the resolution of a model. By using plume modelling, they showed that at relatively high $NO_2$ concentrations, an increase in $NO_2$ results in a decrease of OH and hence longer lifetime of $NO_2$. The corresponding sentence has been changed to (p 14 l 11-13, revised version):

"Moreover, increasing the amount of $NO_2$ at relatively high $NO_2$ concentrations results in a decrease of OH and hence increasing lifetime of $NO_2$ (Valin et al, 2011)."

*Page 19, line 32: It is stated '..as MAX-DOAS instruments usually measure in one azimuthal pointing direction meaning that measurements are performed only on a specific line of sight..'*
*I think this is not true. MAX-DOAS prifile inversions include measurements from many elevation angles (not only from one line of sight).*

*page 19, line 34: It is stated 'This could for example mean that a pollution plume with a horizontal extent on the order of the model resolution and hence showing up in the simulations is missed by the line of sight of the MAX-DOAS instrument'. Again, I think 'line of sight' is not correct here. Also I don't understand why the described plume is missed by the MAX-DOAS measurement.*

The text has been changed to (p 9 l 4-9, revised version, the corresponding paragraph moved to Section 3 as part of the response to referee #2):

"However, in reality, the comparison between horizontal averaging volume of MAX-DOAS and horizontal resolution of the models is much more complicated, as MAX-DOAS instruments usually measure in one azimuthal pointing direction meaning that measurements, though integrated along the line of sight and over several elevation angles, do not necessarily fully represent the amount of $NO_2$ in the three dimensional grid boxes of the model simulations. This could for example mean that a pollution plume with a horizontal extent on the order of the model resolution and hence showing up in the simulations is not observed by the MAX-DOAS instrument."

**Response to anonymous referee #2:**

**We thank referee #2 for taking the time for the review and for providing helpful comments. Our response is given below (assuming that page and line numbers in the referee comments refer to the changes highlighted version of the former manuscript).**

*Despite addressing several of the previous comments and representing a worthwhile effort. if further improved, the revised manuscript fails to address my larger concerns. The language of the writing is informal, the comparisons in the text are qualitative in nature, and the conclusions of the study are too general. In its current form, the manuscript presents a lot of data and a lot of model results, and then lists many factors that could affect both model and observation, but does not provide a satisfying conclusion or an articulate path forward on how future studies might reach conclusions. The first two paragraphs of the Summary and Conclusions section highlight the concerns discussed above:*

*"The reasons for differences between model results and observations found by the comparisons are discussed here in a general sense and need to be further investigated by carrying out additional dedicated model runs in future modelling studies. In general, differences between simulated and retrieved tropospheric NO2 VCDs as well as surface partial columns found in this study could result from model uncertainties in chemistry and meteorology or a combination of both. Moreover, errors related to NOx emission inventories or uncertainties in tropospheric MAX-DOAS retrievals may also contribute to differences between simulated and retrieved values found in this study.*

*"Our analysis shows that in general and on average the model ensemble does well represent tropospheric NO2 amounts observed by MAX-DOAS. However, many points to evaluate arise from the MAX-DOAS based comparisons. Tracking down the reasons for differences between simulations and retrievals and adjusting model runs accordingly (in case of differences caused by errors in simulations rather than uncertainties of the retrievals) could improve model performance substantially."*

*The paper would be improved considerably if the documentation of results was more cohesively and coherently described, removing speculative statements on the cause of discrepancies between models and observations, which only distract from the worthwhile effort to document the discrepancies common to models, between the models and how we could use this information to inform future studies. Currently, the authors prescribe future studies to answer a variety of pressing questions without providing much direction on how the dataset can be used to do so.*

Following the comment by the reviewer, we have removed text which only lists possible reasons for the differences found from the manuscript. Moreover, the 'Summary and Conclusions' Section has been largely rewritten. The results of this study show that future model development needs to concentrate on improving representation of diurnal cycles and associated temporal scalings. This is now pointed out more clearly. Please see the revised version for further details.

*It is not clear why the authors present new analysis and discussion of model resolution effects in the middle of the summary and conclusions section – P19 L22 – P20 L3*

The paragraph has been moved to p 8 l 30 - p 9 l 9 (revised version) and adjusted in response to further review comments.

*Specific comments:*

*P2 L8-22 – Much of this content is general atmospheric chemistry knowledge and does not have much relevance for the rest of the material (e.g., PAN is not mentioned once after this section)*

The text has been removed.

*P3 L26 "In contrast to NO2, NOx cannot be retrieved from MAX-DOAS ... so that these measurements of more interest for air quality than atmospheric chemistry studies." I do not understand why NO2 is of more interest to air quality than atmospheric chemistry studies as stated in the manuscript.*

As NO is rapidly converted to $NO_2$ and vice versa in the troposphere, NOx is usually investigated in atmospheric chemistry studies. $NO_2$ is therefore of more interest for air quality studies. Air quality standards and associated pollution concentration thresholds exist for $NO_2$, but not for NOx.

*P4 31-33 – Consider editing or deleting for clarity*

The sentence has been changed to (p 4 l 3-5, revised version):

"The input to these models is thus consistent and differences in model results are mainly due to different representation of chemical processes, advection, convection, turbulent mixing, wet and dry deposition (Marécal et al., 2015)."

*P5 21 – "More details on specific model setups and scores with respect to surface observations" – Consider editing.*

After consideration, we decided to keep the sentence in the revised version.

*P8 L17-20 – The data screening metrics described here are qualitative. Add quantitative detail.*

Text has been changed to (p 6 l 34 - p7 l 3, revised version) :

"No cloud flags are available for Bremen, De Bilt and OHP. For Bremen, MAX-DOAS retrievals were filtered for patchy cloud situations by comparing radiative forward calculations of $O_4$ to retrieved $O_4$ columns and removing scans with correlations < 0.6. For the other stations, the following data was not considered: (De Bilt) retrievals with fitting residuals > $5 \times 10^{-4}$ and $NO_2$ or $O_4$ fitting errors > 50 %, (Uccle/OHP) retrievals with $NO_2$ fitting rms > 50 % and degrees of freedom for signal < 1."

*P9 - The inclusion of several paragraphs and two equations describing weighted and unweighted average of a quantity is not necessary.*

The text has been shortened and equation 1 of the former version was removed (see first two paragraphs of Section 3 in the revised version).

*P14 L32-35 – This information belongs in the caption of a figure, not in the main body of the text.*

The following sentences were deleted from the main text:

"The number of MAX-DOAS measurements available for each month is given at the top y-axis of each seasonal cycle plot as an indicator of statistical significance. The number of data values is also shown for diurnal and weekly cycle Figures which will be discussed below."

and the last sentence in the caption of Figure 8 (which already contained most information of the sentencences above) has been changed to:

"The number of data values used for calculating average values is shown at the upper x-axis of each plot **as an indicator of statistical significance.**"

*P15 L7 – "There is an overestimation of wintertime values while summertime values are better reproduced by the model ensemble. This may indicate that the model ensemble overestimates production of OH via photolysis of O3 when less light is available, as OH acts as a sink for NO2" The reasoning demonstrated here is representative of reasoning throughout the document – a single*

*process is singled out as the potential source of discrepancy seemingly at random from a multitude of potential causes. I recommend that the authors refrain from speculating on sources of difference without more exhaustive and conclusive analysis. It distracts from the worthwhile pursuit of a thorough documenting of the differences.*

The corresponding text has been deleted from the manuscript.

*P20 L4-6 "A pollution plume and related increase in the time series of tropospheric NO2 VCDs observed by MAX-DOAS would be expected to be reproduced better by model runs with higher horizontal resolution compared to lower resolution runs. The lifetime of NO2 is also expected to increase with model resolution." Please use terms like finer and coarser to describe resolution as lower and higher can be confusing.*

Changed as requested.

[revised manuscript text omitted]